# Modular organization of cerebellar climbing fiber inputs during goal-directed behavior

Shinichiro Tsutsumi[1,2]*, Naoki Hidaka[1,2,3], Yoshikazu Isomura[2,4,5], Masanori Matsuzaki[2,6], Kenji Sakimura[7], Masanobu Kano[1,8]*, Kazuo Kitamura[1,2,3]*

[1]Department of Neurophysiology, Graduate School of Medicine, The University of Tokyo, Tokyo, Japan; [2]CREST, Japan Science and Technology Agency, Saitama, Japan; [3]Department of Neurophysiology, Faculty of Medicine, University of Yamanashi, Yamanashi, Japan; [4]Brain Science Institute, Tamagawa University, Tokyo, Japan; [5]Department of Physiology and Cell Biology, Graduate School of Medical and Dental Sciences, Tokyo Medical and Dental University, Tokyo, Japan; [6]Department of Cellular and Molecular Physiology, Graduate School of Medicine, The University of Tokyo, Tokyo, Japan; [7]Department of Cellular Neurobiology, Brain Research Institute, Niigata University, Niigata, Japan; [8]International Research Center for Neurointelligence (WPI-IRCN), The University of Tokyo Institutes for Advanced Study (UTIAS), The University of Tokyo, Tokyo, Japan

*For correspondence:
stsutsumi@m.u-tokyo.ac.jp (ST);
mkano-tky@m.u-tokyo.ac.jp (MK);
kitamurak@yamanashi.ac.jp (KK)

Competing interests: The authors declare that no competing interests exist.

**Abstract** The cerebellum has a parasagittal modular architecture characterized by precisely organized climbing fiber (CF) projections that are congruent with alternating aldolase C/zebrin II expression. However, the behavioral relevance of CF inputs into individual modules remains poorly understood. Here, we used two-photon calcium imaging in the cerebellar hemisphere Crus II in mice performing an auditory go/no-go task to investigate the functional differences in CF inputs to modules. CF signals in medial modules show anticipatory decreases, early increases, secondary increases, and reward-related increases or decreases, which represent quick motor initiation, go cues, fast motor behavior, and positive reward outcomes. CF signals in lateral modules show early increases and reward-related decreases, which represent no-go and/or go cues and positive reward outcomes. The boundaries of CF functions broadly correspond to those of aldolase C patterning. These results indicate that spatially segregated CF inputs in different modules play distinct roles in the execution of goal-directed behavior.
DOI: https://doi.org/10.7554/eLife.47021.001

## Introduction

Anatomically, the cerebellum is the simplest structure in the brain, comprising a few types of neurons with similar interconnections. The inputs and outputs of the cerebellar cortex are organized into modules (*Voogd and Glickstein, 1998*), wherein spatially defined populations of Purkinje cells (PCs) project to a distinct subdivision of the cerebellar nucleus, which in turn projects to a subnucleus of the inferior olive that provides climbing fiber (CF) inputs to PCs. These modules are called zones or microzones (*Oscarsson, 1979*) and are considered as functional units in the cerebellum. On the other hand, PCs show parasagittal alternating band-like expression of various molecules, including aldolase C/zebrin II (*Brochu et al., 1990*). Each hemi-cerebellum can be subdivided into 14 alternating aldolase C-positive (AldC+) and aldolase C-negative (AldC−) compartments (1+ to 7+), which

are largely congruent with the classical zones (*Sugihara and Shinoda, 2004*; *Voogd and Ruigrok, 2004*).

Accumulating evidence suggests functional difference between AldC+ and AldC− PCs. AldC+ PCs receive enhanced glutamate release by CFs (*Paukert et al., 2010*), but parallel fiber inputs that are similar to those received by AldC− PCs (*Nguyen-Minh et al., 2018*). AldC+ and AldC− PCs show distinct firing properties and plasticity rules (*Zhou et al., 2014*), and differ in interaction between simple and complex spikes (*Tang et al., 2017*). CF inputs to AldC+ and AldC− compartments are separately and precisely synchronized at cellular resolution (*Tsutsumi et al., 2015*). AldC+ and AldC− PCs respond differently to optic flow stimulation in the pigeon cerebellum (*Graham and Wylie, 2012*). Despite these functional implications and the rich anatomical information that is available, surprisingly little is known about the differences in CF inputs across the AldC+ and AldC− compartments and across the mediolateral axis of the cerebellar lobules during behavior.

To address this issue systematically, we expressed the genetically encoded calcium indicator GCaMP6f (*Chen et al., 2013*) in the cerebellar hemisphere folium Crus II of Aldoc-tdTomato mice (*Tsutsumi et al., 2015*), which express the red fluorescent protein tdTomato in AldC+ PCs, and conducted two-photon calcium imaging of PC dendrites of awake mice performing a lick/no-lick task. We found that the proportion of AldC+ PCs showing CF-dependent dendritic $Ca^{2+}$ signals (CF signals) is scaled with licking behavior and higher than that of AldC− PCs. CF signals in the medial Crus II were prevalent during licking trials and scaled with licking behavior, whereas those in the lateral Crus II showed equal prevalence in all trial types or were pronounced after no-go cues. We found anticipatory decreases in population CF signals toward sensory cues in a medial AldC+ compartment, which were correlated with the initiation of licking, and in a lateral AldC− compartment, which were related to delay in lick initiation. Early increases of CF signals after sensory cues in the medial Crus II were correlated with the initiation of licking, quicker lick response, faster licking, and the presence of go cues, whereas those in lateral AldC− compartments were correlated with the presence of no-go cues. Subsequent increases in CF signals in the medial Crus II reflected faster licking. Later decreases in CF signals in the lateral Crus II and increases in a medial AldC− compartment after licking represented positive reward outcomes. These results suggest distinct contributions of spatially organized CF inputs to the Crus II to sensorimotor behavior.

## Results

### The cerebellar Crus II contributes to an auditory lick/no-lick task

We trained mice to perform a lick/no-lick auditory discrimination task (*Figure 1A*), which required the initiation or suppression of licking in response to different auditory cues. We reasoned that the task requires sensorimotor functions beyond motor correction or adaptive control of reflex, and therefore efficiently reveals functional differences across AldC compartments during behavior. In this task, mice had to respond to a high-frequency tone (go cue) with a lick to receive a water reward (hit; *Figure 1A*). A lick after a low-frequency tone (no-go cue) was punished with a timeout (4.5 s; false alarm or FA). No licking after a go cue (miss) or a no-go cue (correct rejection or CR) was neither rewarded nor punished. The inter-trial interval was set to 6 s, but a subsequent trial was delayed for 1 s after the cessation of continued licking. Well-trained mice almost completely withheld licking after a no-go cue (*Figure 1B*) and cut off protracted licking, yielding a constant inter-trial interval (hit, CR, and miss trials: 6.23 ± 0.58 s; FA trials: 10.98 ± 1.65 s; mean ± s.d., *n* = 17 mice). Licking in hit trials was shorter in latency, higher in frequency, longer in duration, and more regular than licking in FA trials (*Figure 1C and D*, and *Figure 1—figure supplement 1C*). The performance of mice quickly improved to an expert level, defined as fraction of correct trials >80%, within 3–6 sessions (4.18 ± 0.27 sessions, mean ± s.e.m., *n* = 17 mice; *Figure 1E*) by first increasing the hit rate, and then increasing the CR rate (*Figure 1—figure supplement 1A and B*). Licking behavior in hit trials remained relatively unchanged (*Figure 1F and G*, and *Figure 1—figure supplement 1D*). By contrast, the frequency of FA licking robustly decreased and became irregular (*Figure 1H and I*, and *Figure 1—figure supplement 1E*). Therefore, mice learned to suppress their licking in response to a no-go cue, while preserving their licking in response to a go cue.

We assessed the contribution of the cerebellar folium Crus II, a region associated with licking (*Bryant et al., 2010*; *Welsh et al., 1995*), to this go/no-go behavior by injecting a $GABA_A$ receptor

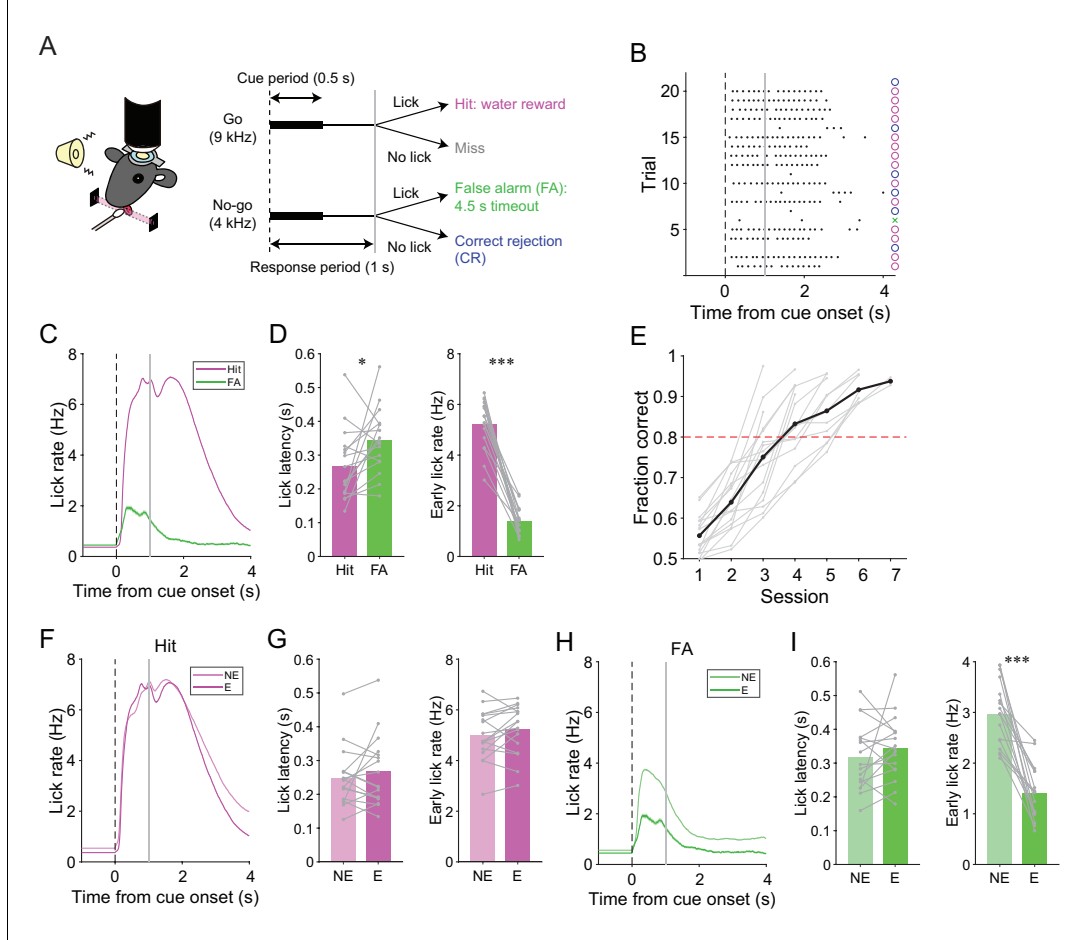

**Figure 1.** A lick/no-lick auditory discrimination task. (**A**) Left, schematic diagram of a mouse performing the task under a two-photon microscope. Right, task structure. (**B**) Performance of an expert mouse in a representative session. Black dots represent licks. A vertical dotted line represents cue onset, and a gray line indicates the end of the response window. Circles and a cross represent correct trials and an incorrect trial, respectively. The color scheme is the same as that in panel (**A**). (**C**) Trial-averaged lick rate for hit and FA trials pooled across all trials in expert performance sessions. (**D**) Mouse-averaged lick latency (left) and lick rate within 1 s from cue onset (right) for hit and FA trials in expert performance sessions. Gray dots and lines represent individual mice (*n* = 17). (**E**) Learning curve for all mice (*n* = 17). Black dots and a line represent an average. A red dashed line represents a threshold for expert performance level (fraction correct = 0.8). (**F**) Trial-averaged lick rate for hit trials pooled across expert (E) and non-expert (NE) performance sessions. (**G**) Mouse-averaged lick latency (left) and lick rate within 1 s from cue onset (right) for hit trials in expert (E) and non-expert (NE) performance sessions. (**H**) Same as panel (**F**) but for FA trials. (**I**) Same as panel (**G**) but for FA trials. (**C, F, H**) Lines and shadings represent means ± s.e.m. (**D, G, I**) Paired *t*-test: *p<0.05, ***p<0.001.

DOI: https://doi.org/10.7554/eLife.47021.002

The following source data and figure supplements are available for figure 1:

**Source data 1.** Datasets used to create *Figure 1*.
DOI: https://doi.org/10.7554/eLife.47021.007
**Figure supplement 1.** Analyses of task performance and licking behavior.
DOI: https://doi.org/10.7554/eLife.47021.003
**Figure supplement 1—source data 1.** Datasets used to create *Figure 1—figure supplement 1*.
DOI: https://doi.org/10.7554/eLife.47021.004
**Figure supplement 2.** Effects of muscimol on mouse behavior.
DOI: https://doi.org/10.7554/eLife.47021.005
**Figure supplement 2—source data 1.** Datasets used to create *Figure 1—figure supplement 2*.
DOI: https://doi.org/10.7554/eLife.47021.006

agonist, muscimol, at the center of the left Crus II just before the behavioral sessions in fully trained wild-type mice ($n$ = 5; after total 6 ± 1 sessions and 1827 ± 278 trials of the lick/no-lick task; mean ± s.d.). The muscimol injection decreased the hit rate compared with the saline control (*Figure 1—figure supplement 2A and B*). Moreover, licking in hit trials was delayed and slower in muscimol injection sessions (*Figure 1—figure supplement 2C and D*), and licking in FA trials was also delayed by muscimol injection (*Figure 1—figure supplement 2E and F*). These results indicate that the Crus II contributes to the proper initiation and speed of conditioned licking.

## Correspondence between task-related CF signals and AldC expression

We used the Aldoc-tdTomato transgenic mouse line to explore systematically functional differences in CF inputs to AldC+ and AldC− compartments at single-cell resolution during the task. In this mouse line, AldC expression can be visualized in vivo as red tdTomato fluorescence (*Figure 2A and B*, left); we have previously shown by immunohistochemistry that the expression of tdTomato precisely corresponds to that of AldC (*Tsutsumi et al., 2015*). We virally expressed GCaMP6f in PCs of the left Crus II, and performed two-photon calcium imaging from PC dendrites at every boundary of AldC expression to simultaneously monitor CF-dependent dendritic Ca$^{2+}$ signals (CF signals) in adjacent AldC+ and AldC− compartments (*Figure 2B*, middle), as previously described (*Tsutsumi et al., 2015*). We extracted 10–20 (13.7 ± 5.7; mean ± s.d.) dendritic regions of interest (ROIs) from single AldC compartments ($n$ = 472 compartments; *Figure 2B*, right). We observed global calcium increases in the dendrites of each PC, which reflect complex spikes that represent responses of PCs to CF inputs (*Kitamura and Häusser, 2011*; *Tsutsumi et al., 2015*) that are triggered by auditory cues and licking (*Figure 2C*). CF signals were observed immediately after the cue onset and were robust during hit trials in AldC+ PCs (5+ AldC compartment; *Figure 2D and E*), whereas much smaller responses were observed in AldC− PCs (5− AldC compartment; *Figure 2D and E*). This response pattern was clearly separated at the boundary of AldC expression (*Figure 2D*).

To examine whether the boundaries of functional difference in CF signals correspond to those of AldC expression at microzone level, we averaged the response probabilities of individual ROIs located within 100 μm bins from the AldC boundaries. Differences in response probability were noted across the AldC boundaries for either or all of hit, FA, and CR trials (*Figure 2—figure supplement 1*), indicating that the AldC expression boundaries have a close (≦ 100 μm) spatial correspondence with the CF functional boundaries.

To further explore whether the boundaries of functional difference in CF signals correspond to those of AldC expression at the cellular level, we performed principal component analysis on whole traces in an imaging session from all ROIs irrespective of the AldC expression profile in each imaging field of view, and segregated these ROIs using $k$-means clustering performed on the basis of the first three principal components (PrC1, PrC2, and PrC3; *Figure 2—figure supplement 2A*). Cluster number was chosen on the basis of the largest average silhouette value (*Rousseeuw, 1987*), which represents how well each cluster is separated from the others. Cluster number in single field of view was between 2 and 5, and it was higher at a medial AldC boundary than at a lateral boundary (7+/6− vs. 5+/5a−; *Figure 2—figure supplement 4A*). Silhouette values ranged from 0.7 to 0.9, indicating good separation of clusters for some, but not all, of the boundaries (*Figure 2—figure supplement 4B*). Indeed, each cluster showed distinct task-related CF signals (*Figure 2—figure supplement 3*), supporting their functional separation. Then we projected the clustering results to the tdTomato reference images and calculated average coincidence rate representing the percentage of ROIs showing the same AldC expression profile within each cluster (*Figure 2—figure supplement 2B*). High coincidence value roughly corresponded to good separation of correlation matrices across ROIs in individual imaging sessions (*Figure 2—figure supplement 2C*). Mean coincidence rates across animals were as high as 90% in 7+/6−, 5−/5+, and 5+/5a− boundaries (*Figure 2—figure supplement 4C*), indicating that unbiased clustering of ROIs on the basis of their responses corresponds to separation using AldC expression at cellular resolution at these boundaries, but more broadly tuned in the other boundaries.

On average, the percentage of ROIs that respond during the early time window (0–0.5 s from cue onset) in AldC+ compartments was higher for licking trials (hit and FA) than for non-licking trials (CR), and scaled with lick rate (hit > FA; *Figure 2F*). The percentage of responsive ROIs in AldC+ compartments during hit trials was significantly higher than that in AldC− compartments

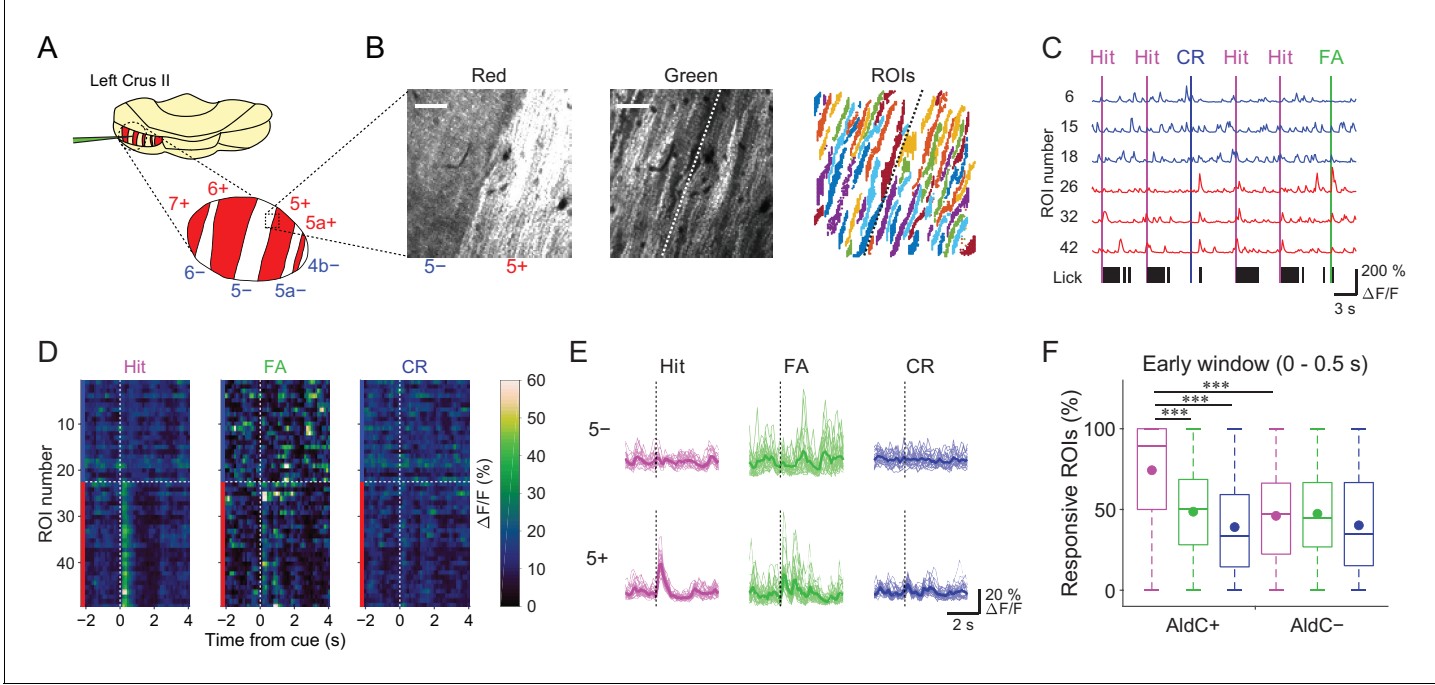

**Figure 2.** Differences in task-related CF signals between AldC+ and AldC− compartments. (**A**) Schematic diagram of virus injection into the left cerebellar folium Crus II of Aldoc-tdTomato mice. Inset, magnified view of the Crus II. Red stripes represent AldC+ compartments. (**B**) Left, tdTomato fluorescence image at a 5−/5+ boundary. Middle, mean image of a GCaMP6f video at the same field of view. Right, extracted ROIs representing PC dendrites that are pseudocolored, in the same field of view. Scale bars, 40 μm. (**C**) Representative calcium traces from dendrites in panel (**B**) and their corresponding lick activities (black bars). Colored vertical lines represent the onsets of sensory cues for each trial type. (**D**) Trial-averaged calcium traces of single ROIs from the data in panel (**C**). (**E**) Single ROI trial-averaged traces for 5− and 5+ compartments from the data in panel (**C**). Thick lines represent ROI-averaged traces. (**F**) Percentage of responsive ROIs in a compartment per session for each trial type pooled across all AldC+ or AldC− compartments (n = 17 mice, 79 imaging sessions). Colored circles represent means. Colors represent trial types (hit, magenta; FA, green; CR, blue). Two-way ANOVA on ranks with repeated measures followed by post-hoc Tukey's test: ***p<0.001.

DOI: https://doi.org/10.7554/eLife.47021.008

The following source data and figure supplements are available for figure 2:

**Source data 1.** Datasets used to create *Figure 2*.
DOI: https://doi.org/10.7554/eLife.47021.015
**Figure supplement 1.** Broad correspondence between functional boundaries and AldC expression boundaries at the microzone level.
DOI: https://doi.org/10.7554/eLife.47021.009
**Figure supplement 1—source data 1.** Datasets used to create *Figure 2—figure supplement 1*.
DOI: https://doi.org/10.7554/eLife.47021.010
**Figure supplement 2.** Broad correspondence between functional boundaries and AldC expression boundaries at cellular resolution.
DOI: https://doi.org/10.7554/eLife.47021.011
**Figure supplement 3.** Task-related CF signals in individual clusters.
DOI: https://doi.org/10.7554/eLife.47021.012
**Figure supplement 4.** Summary of correspondence between functional boundaries and AldC expression.
DOI: https://doi.org/10.7554/eLife.47021.013
**Figure supplement 4—source data 1.** Datasets used to create *Figure 2—figure supplement 4*.
DOI: https://doi.org/10.7554/eLife.47021.014

(*Figure 2F*), suggesting that CF inputs to AldC+ compartments are more relevant to motor behavior than those to AldC− compartments.

## Mediolateral distinction in sensory and motor-related CF signals

Previous reports on anatomical CF projections in rodents (*Sugihara and Shinoda, 2004*; *Sugihara and Quy, 2007*) suggested functional differences across the mediolateral axis in the cerebellar hemisphere. We took advantage of our mouse line to precisely locate the individual AldC+

and AldC− compartments on the basis of their characteristic widths and relative locations (see 'Materials and methods') (*Tsutsumi et al., 2015*), which enabled us to explore the differences in functional CF inputs along the mediolateral axis. We found differences in task-related CF signals in laterally and medially located AldC+ compartments (*Figure 3A and B*). First, CF signals in a lateral AldC+ compartment (7+) during hit trials were precisely time-locked to sensory cues followed by suppression (*Figure 3B*, top), whereas those in a medial AldC+ compartment (5a+) persisted long after the cues (*Figure 3B*, bottom). Second, task-related CF signals were distinct across AldC expression boundaries at lateral AldC compartments (7+/6−), whereas there was less difference across medial AldC compartments (5a+/4b−; *Figure 3B*). Finally, task-related CF signals were graded in extent depending on lick rate (hit > FA) in the medial AldC compartments (5a+ and 4b−), whereas there was a distinct difference between responses to go and no-go cues in the lateral AldC compartment (7+; *Figure 3B*).

To investigate whether there is a systematic functional difference along the medio-lateral axis in the Crus II, we characterized CF signals across all boundaries (*Figure 3—figure supplement 1A and B*), aligned task-related CF signals in individual compartments and sorted them from lateral to

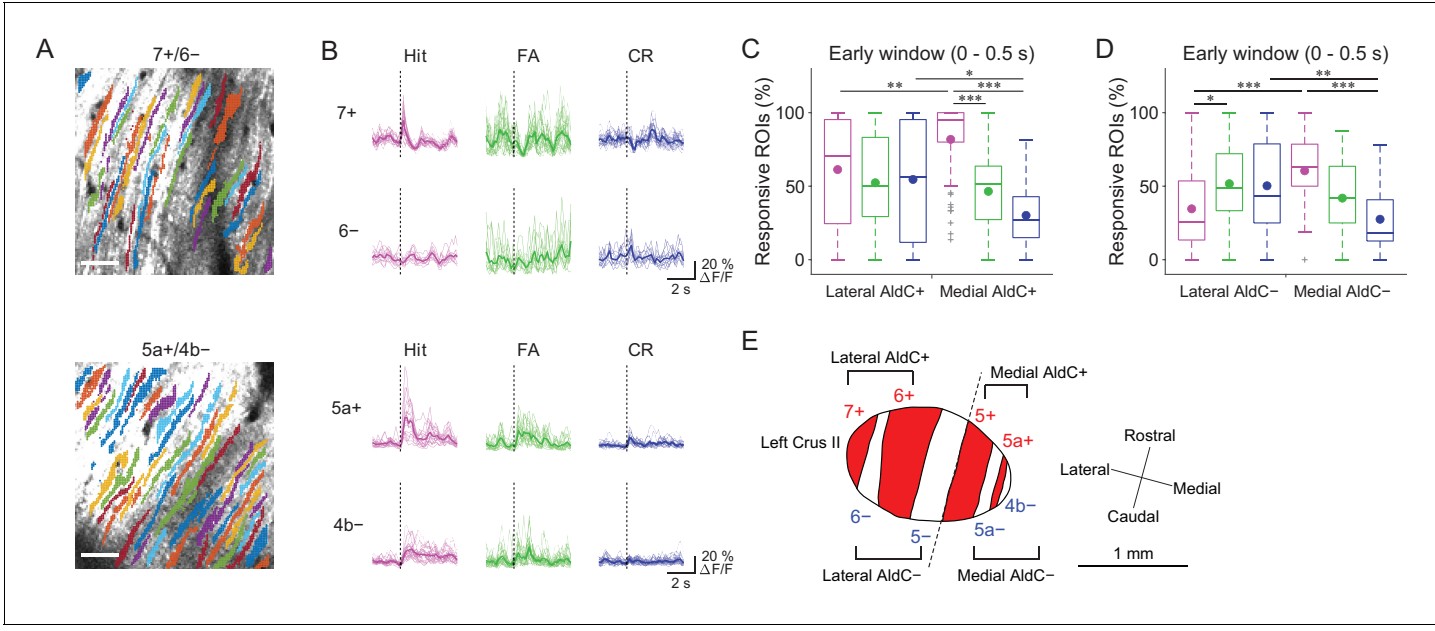

**Figure 3.** Difference in CF signals between the lateral and medial Crus II. (A) Extracted ROIs representing PC dendrites that are pseudocolored overlaid on the tdTomato image around a 7+/6− (top) and a 5a+/4b− (bottom) boundary. Scale bars, 40 µm. (B) Single ROI trial-averaged traces for 7+ and 6− (top) and 5a+ and 4b− (bottom) compartments from the data in panel (A). Thick lines represent ROI-averaged traces. (C) Percentage of responsive ROIs during the early response window (0 to 0.5 s from cue onset) in an AldC compartment per session for each trial type pooled across all the lateral AldC+ or medial AldC+ compartments (*n* = 12 and 13 mice, 29 and 50 imaging sessions, respectively). Colored dots represent means. Colors represent trial types (hit, magenta; FA, green; CR, blue). (D) Same as panel (C) but for all the lateral AldC− or medial AldC− compartments (*n* = 15 and 9 mice, 44 and 35 imaging sessions, respectively). (E) Schematic diagram of the left Crus II showing a functional grouping of AldC compartments. The dashed line represents a boundary between the lateral and medial Crus II. (C, D) Two-way ANOVA on ranks with repeated measures followed by post-hoc Tukey's test: *p<0.05, **p<0.01, ***p<0.001.

DOI: https://doi.org/10.7554/eLife.47021.016

The following source data and figure supplements are available for figure 3:

**Source data 1.** Datasets used to create *Figure 3*.
DOI: https://doi.org/10.7554/eLife.47021.020

**Figure supplement 1.** Mediolateral separation of task-related CF signals in the Crus II.
DOI: https://doi.org/10.7554/eLife.47021.017

**Figure supplement 1—source data 1.** Datasets used to create *Figure 3—figure supplement 1*.
DOI: https://doi.org/10.7554/eLife.47021.018

**Figure supplement 2.** Difference in secondary CF signals between the lateral and medial Crus II.
DOI: https://doi.org/10.7554/eLife.47021.019

medial (*Figure 3—figure supplement 1C*). On the basis of these task-related CF signals in individual compartments, the lateral and medial Crus II were separable at the boundary of 5−/5+ (*Figure 3—figure supplement 1C*), where complete separation of the functional clusters and the correlation matrix was observed (*Figure 2—figure supplement 2C*). Therefore, we grouped 7+ and 6+ as lateral AldC+, 6− and 5− as lateral AldC−, 5+ and 5a+ as medial AldC+, and 5a− and 4b− as medial AldC− compartments (*Figure 3E*).

To show the mediolateral difference in task-related CF signals quantitatively, we compared percentages of ROIs that responded during the early time window (0 to 0.5 s from cue onset) in the lateral AldC+ versus medial AldC+ compartments, and in the lateral AldC− versus medial AldC− compartments, given the functional difference across the AldC+ and AldC− compartments (*Figure 2*). We found that the percentage of ROIs responding during hit trials was higher for the medial AldC compartments than for the lateral AldC compartments, irrespective of AldC expression profile (*Figure 3C and D*). The percentage was graded depending on the lick rate in the medial AldC compartments (hit > FA; *Figure 3C and D*), whereas that in the lateral AldC compartments was equal for all trial types (*Figure 3C*) or pronounced in FA trials (*Figure 3D*). The percentage of responding ROIs for CR trials was higher for the lateral than for the medial AldC compartments, again irrespective of AldC expression (*Figure 3C and D*). In addition, given the observed difference in the temporal structure of CF signals across the lateral and medial AldC compartments (7+, hit vs. 5a+, hit; *Figure 3B*), we extended the same analysis to secondary increases in CF signals (secondary time window; 0.5 to 1 s from cue onset; *Figure 3—figure supplement 2*). Differences in CF signals similar to those observed in early time window were observed across the lateral and medial AldC compartments: there was a higher percentage of hit-responsive ROIs, which were scaled with licking activity, in the medial AldC compartments (*Figure 3—figure supplement 2*). These results suggest that CF inputs to the lateral AldC compartments report task onsets regardless of motor behavior, whereas those to the medial AldC compartments reflect motor behavior.

## Temporal organization of CF signals during a go/no-go task

As noted above, close inspection of task-related CF signals in individual AldC compartments revealed several temporal components. We found slight decreases in CF signals during the pre-cue (−1 to 0 s from cue) period in the 5+ compartment (effect size = 0.18; *Supplementary file 1*), early increase (0 to 0.5 s from cue) in CF signals in all the AldC compartments, and secondary increase (2nd; 0.5 to 1 s from cue) in all the medial AldC compartments (*Figure 4—figure supplement 1*). These pre-cue decreases and early and secondary increases in CF signals can co-exist, even in single dendrites at single trials (*Figure 4—figure supplement 2A and B*). Alignment of CF signals at lick onset in task-related and non-task-related lick bouts revealed a decrease in CF signals in the lateral AldC compartments and an increase in the 5a− compartment at late timing (1.2 to 2.2 s from lick onset) during hit trials (*Figure 4—figure supplement 3*). Therefore, we separately analyzed CF signals within these time windows to investigate their behavioral relevance.

## Anticipatory decreases in CF signals are correlated with motor initiation

Pre-cue (−1 to 0 s from cue onset) decreases in CF signals were detected when the single-trial-population-averaged $Ca^{2+}$ trace within an AldC compartment is below mean − 0.5 s.d. of the baseline (see 'Materials and methods'), representing a relative reduction of CF inputs. Pre-cue decreases in population CF signals in the 5+ compartment in single trials were reliably captured by this method (pre-cue (↓); *Figure 4A*).

Given the involvement of cerebellar activity in motor preparation (*Chabrol et al., 2019*; *Gao et al., 2018*), we reasoned that these decreases in CF signals just before the presentation of a cue could be related to the preparation and/or initiation of licking. To test this possibility, we built a generalized linear model with mixed effects (GLMM) to fit single trial population averaged ΔF/F in single AldC compartments during the pre-cue time window, using the presence (1) or absence (0) of licking in the trial (see Materials and methods). We found that the pre-cue decreases in CF signals in the 5+ compartment (medial AldC+) were significantly correlated with the initiation of licking in the trial (p=0.04; *Figure 4B*), indicating their role in motor preparation.

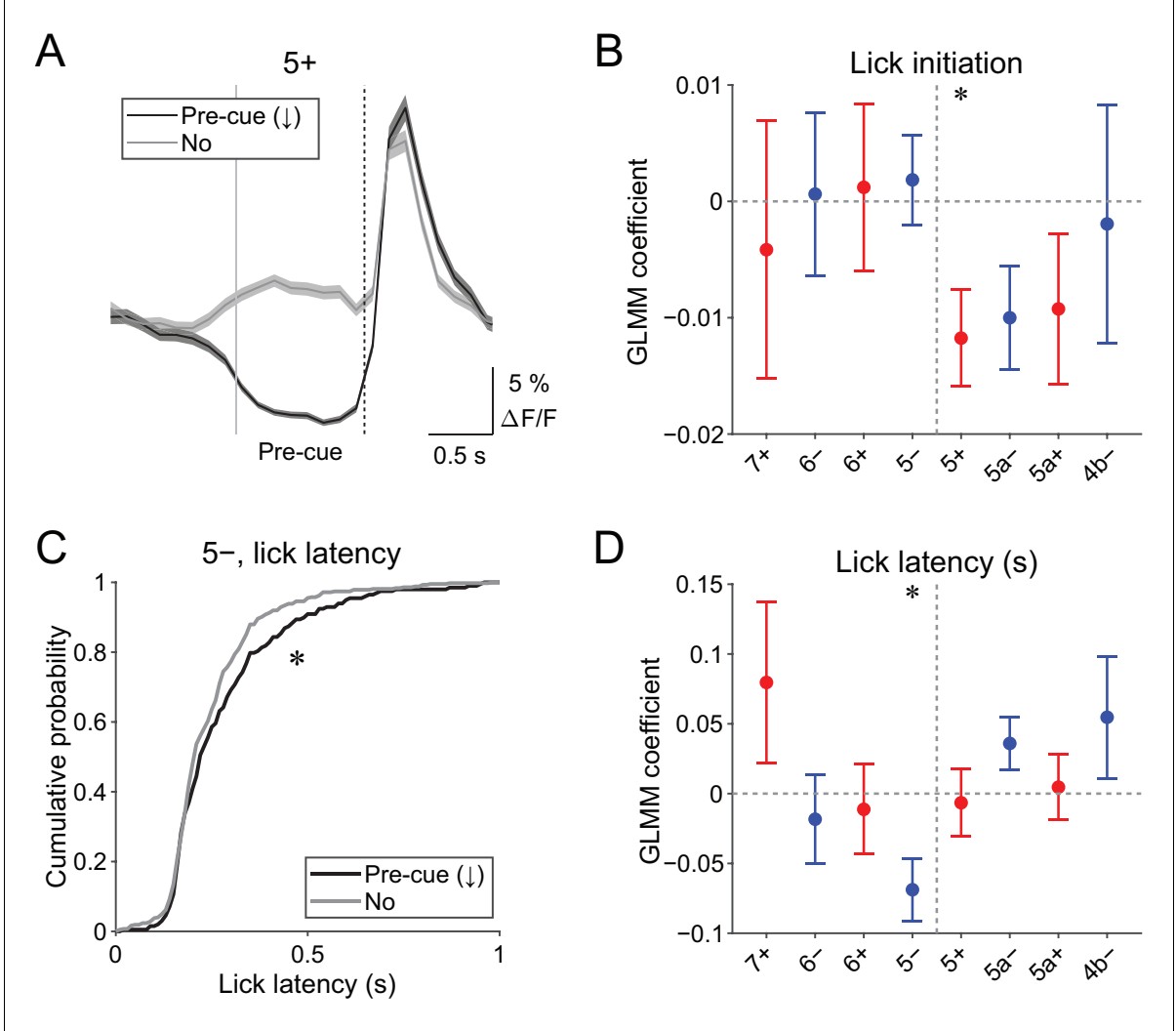

**Figure 4.** Behavioral relevance of pre-cue decrease in CF signals. (A) Single trial calcium traces in a 5+ compartment with (pre-cue (↓); black) or without (no; gray) decreases in CF signals, showing difference in the pre-cue response window (–1 to 0 s from the cue onset; indicated by gray and dashed vertical lines). A dashed vertical line represents the cue onset. Thick lines and shadings represent mean ± s.e.m.. (B), Coefficients of GLMM fit for single trial pre-cue ΔF/F in each AldC compartment by the presence (1) or absence (0) of lick initiation in the following trial (n = 5, 6, 10, 12, 12, 10, 3, and two mice; 393, 605, 649, 1078, 1281, 1051, 664, and 253 trials for 7+, 6−, 6+, 5−, 5+, 5a−, 5a+, and 4b− compartments, respectively). (C), Cumulative distribution of lick latency from cue onsets during licking trials (hit and FA) with (pre-cue (↓); black) or without (no; gray) pre-cuedecreases in CF signals in a 5− compartment. (D) Coefficients of GLMM fit for single trial pre-cue ΔF/F in each AldC compartment before licking trials by latency of lick initiation in the trial (n = 5, 6, 10, 12, 12, 10, 3, and two mice; 225, 346, 384, 621, 722, 586, 353, and 131 trials for 7+, 6−, 6+, 5−, 5+, 5a−, 5a+, and 4b− compartments, respectively). (B, D), Colored dots and error bars represent mean ± s.e.m. Red and blue colors correspond to AldC+ and AldC− compartments, respectively. ANOVA on GLMM fit: *p<0.05. (C) KS test: *p<0.05.
DOI: https://doi.org/10.7554/eLife.47021.021

The following source data and figure supplements are available for figure 4:

**Source data 1.** Datasets used to create *Figure 4*.
DOI: https://doi.org/10.7554/eLife.47021.026
**Figure supplement 1.** Task-related CF signals in all the compartments.
DOI: https://doi.org/10.7554/eLife.47021.022
**Figure supplement 1—source data 1.** Datasets used to create *Figure 4—figure supplements 1* and *3*.
DOI: https://doi.org/10.7554/eLife.47021.023
**Figure supplement 2.** Single ROI single trial CF signals.
DOI: https://doi.org/10.7554/eLife.47021.024
**Figure supplement 3.** CF signals at late timing during licking trials and nontask licking epochs.
DOI: https://doi.org/10.7554/eLife.47021.025

The other possibility is that anticipatory decreases in CF signals represent a decrease in animal's engagement in the task. To address this, we compared the latency of first lick during licking trials (hit and FA) with or without pre-cue decrease in CF signals in an AldC compartment (*Figure 4C*). We found that lick initiation was significantly delayed when there were pre-cue decreases in CF signals in the 5− compartment in the trial (p=0.03; *Figure 4C*). We also built a separate GLMM to fit single trial population averaged ΔF/F in single AldC compartments during the pre-cue time window using lick latency in the trial, and found that pre-cue decreases in CF signals in the 5− compartment were significantly correlated with longer lick latency (p=0.02; *Figure 4D*), indicating the role of CF inputs to this compartment in an animal's attention to the task.

## Early increases in CF signals are correlated with licking behavior and context discrimination

Early (0 to 0.5 s from cue) increases in CF signals were the most prominent feature in all of the AldC compartments that we observed (*Figure 4—figure supplement 1*), which was reliably captured on a single trial basis (*Figure 5A* and *Figure 4—figure supplement 2*). We assessed the behavioral relevance of these signals by examining the following three points.

First, to differentiate sensory and motor components in the early CF signals, we used GLMM to fit single trial population averaged ΔF/F in single AldC compartments during the early response window using the presence (1) or absence (0) of lick in the trial and go (1) or no-go cue (0) in the trial. We found that early increases in CF signals in the 6−, 5+, and 5a− compartments were significantly correlated with lick initiation (p=0.03, 0.01, and 0.02; *Figure 5B*, left). Go and no-go cues were oppositely represented in the lateral and medial AldC compartments: early increases in CF signals in the 6− and 5− compartments (lateral AldC−) were significantly correlated with no-go cues (p=0.0008 and 0.01), whereas those in the 5+, 5a+, and 4b− compartments (medial AldC) were significantly correlated with go cues (p=$2 \times 10^{-7}$, $7 \times 10^{-5}$, and 0.02; *Figure 5B*, right).

Second, we examined the role of CFs in motor timing (*Llinás, 2011*; *Welsh et al., 1995*; *Welsh, 2002*) in our task. We extracted licking trials (hit and FA) with or without early CF signals in an AldC compartment and compared the distribution of lick latency to the cue onset (*Figure 5C*). We found that the presence of early CF signals in the 5+ compartment was significantly correlated with shorter lick latency (p=$3 \times 10^{-5}$; *Figure 5C*). Indeed, GLMM fitting of single trial population averaged ΔF/F in single AldC compartments during licking trials using lick latency revealed significantly negative correlation in the 5+, 5a−, and 4b− compartments (medial AldC); lick latency was shorter with more CF signals in these AldC compartments (p=0.02, 0.0008, and 0.05; *Figure 5D*).

Last, we addressed whether the early CF signals are correlated with licking frequency (*Bryant et al., 2010*). We compared instantaneous lick rate in the licking trials with or without early CF signals in an AldC compartment, and found that the presence of early CF signals in the 5+ compartment was significantly correlated with faster lick rate (p=$4 \times 10^{-6}$; *Figure 5E*). GLMM fitting of single trial population averaged ΔF/F in single AldC compartments during licking trials (as in *Figure 5D*) revealed a significant positive correlation in the 5+ and 5a+ compartments (medial AldC +): lick rate was faster with more CF signals in these compartments (p=0.01 and 0.02; *Figure 5F*).

In summary, early increases in CF signals reflect lick initiation and context discrimination, and licking is shorter in latency and faster when CF signals show early increases in the medial AldC compartments.

## Secondary increases in CF signals reflect speed of licking

Secondary increases in CF signals (0.5 to 1 s from the cue) were noted during hit trials in all the medial compartments (*Figure 4—figure supplement 1*), and these signals were reliably captured on a single trial basis (*Figure 6A* and *Figure 4—figure supplement 2*). Since these signals were observed after lick initiation, we reasoned that they could reflect parameters of already initiated licking such as lick rate. Therefore, we compared instantaneous lick rate in the licking trials with or without secondary increases in CF signals in an AldC compartment (*Figure 6B*). We found that the presence of secondary increases in CF signals in the 5a− compartment was significantly correlated with faster licking (p=0.02; *Figure 6B*). GLMM fitting of single trial population averaged ΔF/F in single AldC compartments using the lick rate of the first lick bout in the single licking trial revealed significantly positive correlation in the 5+, 5a−, and 4b− compartments (medial AldC); lick rate was

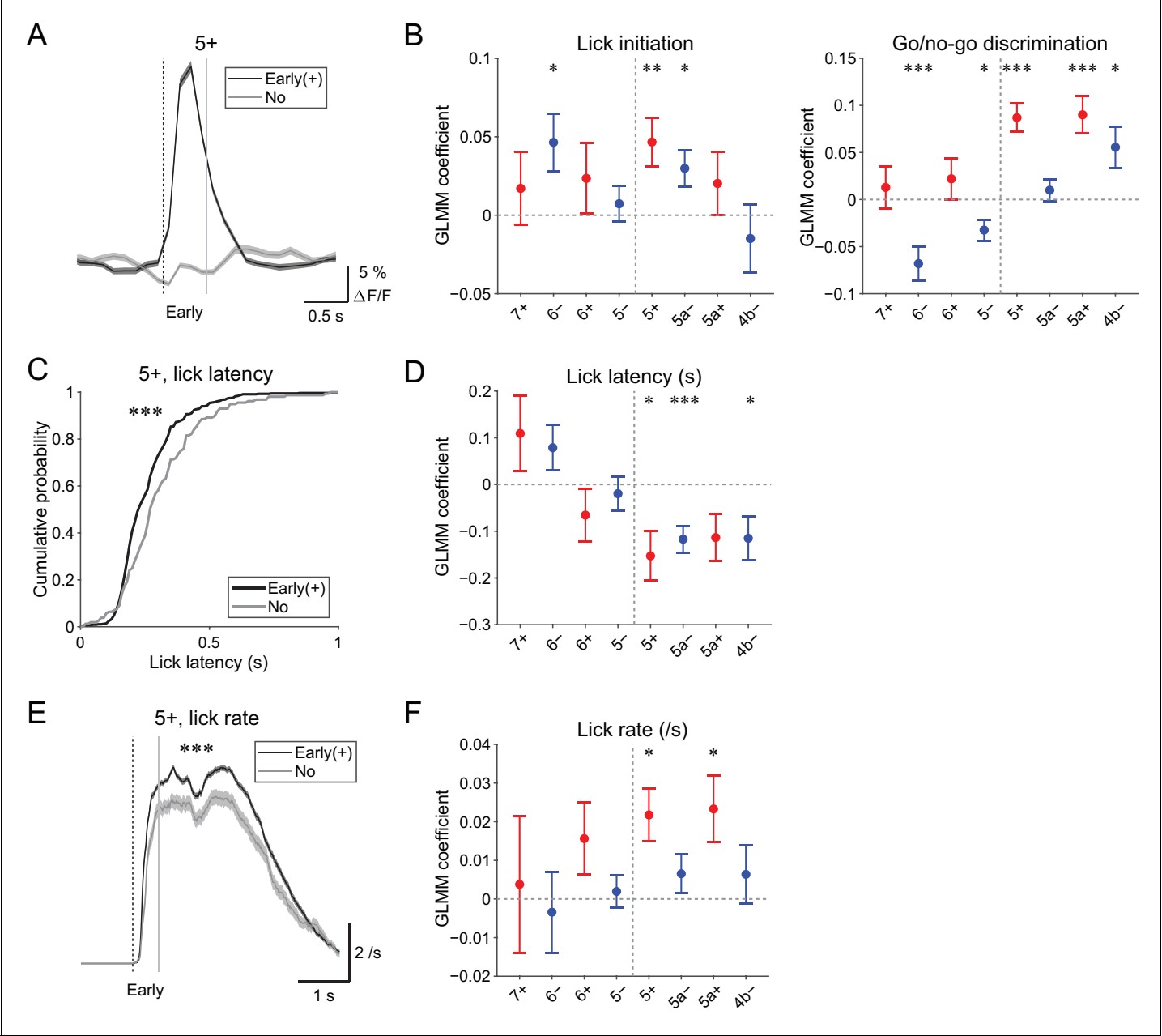

**Figure 5.** Behavioral relevance of early increase in CF signals. (A) Single trial calcium traces in a 5+ compartment with (early (+); black) or without (no; gray) increases in CF signals showing a difference in the early response window (0 to 0.5 s from cue onset). (B) Left, coefficients for lick initiation from GLMM fit for single trial ΔF/F during the early response window in each AldC compartment during all trials by the presence (1) or absence (0) of lick initiation in the trial and go (1) or no-go (0) cue in the trial (n = 5, 6, 10, 12, 12, 10, 3, and two mice; 393, 605, 649, 1,078, 1,281, 1,051, 664, and 253 trials for 7+, 6−, 6+, 5−, 5+, 5a−, 5a+, and 4b− compartments, respectively). Right, same as left, but coefficients for go/no-go cue. (C) Cumulative distributions of lick latency from cue onset during licking trials (hit and FA) with (early (+); black) or without (no; gray) early increases in CF signals in a 5 + compartment. (D) Coefficients for latency of lick initiation from GLMM fit for trial-by-trial ΔF/F during the early response window in each compartment during licking trials in the trial by the lick latency (n = 5, 6, 10, 12, 12, 10, 3, and two mice; 225, 346, 384, 621, 722, 586, 353, and 131 trials for 7+, 6−, 6+, 5−, 5+, 5a−, 5a+, and 4b− compartments, respectively). (E), Instantaneous lick rate during licking trials (hit and FA) with (early (+); black) or without (no; gray) early increases in CF signals in a 5+ compartment (n = 12 mice, 565 and 157 trials, respectively). (F) Same as panel (D), but coefficients are for the lick rate of the first lick bout in the trial. (A, E) Early response window is indicated by a dashed vertical line representing the cue onset and a gray vertical line. Thick lines and shadings represent mean ± s.e.m. (B, D, F) Colored dots and error bars represent mean ± s.e.m. Red and blue colors correspond to the AldC+ and AldC− compartments, respectively. ANOVA on GLMM fit: *p<0.05, **p<0.01, ***p<0.001. (C) KS test: ***p<0.001. (E) One-way ANOVA with repeated measures: ***p<0.001.

DOI: https://doi.org/10.7554/eLife.47021.027

*Figure 5 continued on next page*

*Figure 5 continued*

The following source data is available for figure 5:

**Source data 1.** Datasets used to create *Figure 5*.

DOI: https://doi.org/10.7554/eLife.47021.028

faster with more CF signals in these AldC compartments (p=0.04, 0.04, and 0.008; *Figure 6C*), indicating the importance of the medial Crus II for speed of licking.

## Late changes in CF signals represent positive reward outcomes

We discovered decreases in CF signals at late timing after lick initiation (1.2 to 2.2 s from lick onset) during hit trials in lateral AldC compartments (*Figure 4—figure supplement 3*), which were reliably captured on a single trial basis (*Figure 7A*). On the contrary, increases in CF signals during this time period were observed in the 5a− compartment (*Figure 7B* and *Figure 4—figure supplement 3*). Because this late timing corresponds to the offset of reward delivery, we reasoned that these signals could represent trial outcomes. Therefore, we fitted single trial population averaged ΔF/F in single AldC compartments during this late window by the presence (1) and absence (0) of reward in the trial using GLMM. We found that decreases in CF signals in all the lateral AldC compartments and in the 5+ compartment, as well as increases in the 5a− compartment, were significantly correlated with the presence of reward (7+, p=0.0002; 6−, p=3 × $10^{-12}$; 6+, p=1 × $10^{-12}$; 5−, p=2 × $10^{-7}$; 5+, p=0.0008; 5a−, p=3 × $10^{-5}$; *Figure 7C*), indicating that these signals reflect positive reward outcomes.

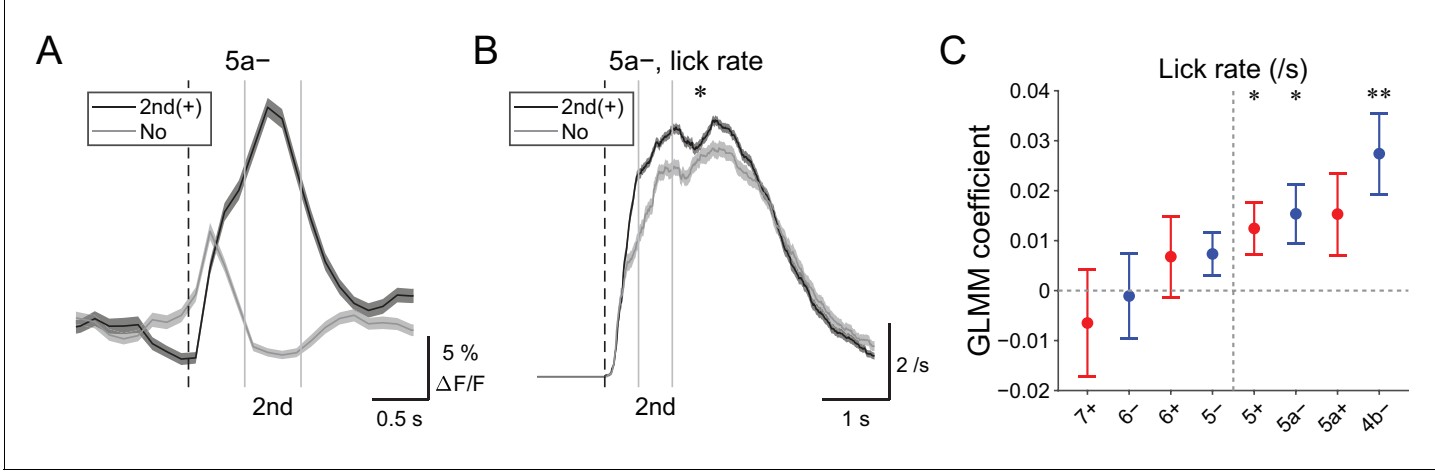

**Figure 6.** | Behavioral relevance of secondary increase in CF signals. (**A**) Single trial calcium traces in a 5a− compartment with (2nd (+); black) or without (no; gray) increases in CF signals showing difference during the secondary response window (0.5 to 1 s from cue onset). (**B**) Instantaneous lick rate during licking trials (hit and FA) with (2nd (+); black) or without (no; gray) increases in CF signals during the secondary response window in a 5a− compartment (*n* = 10 mice, 349 and 237 trials, respectively). (**C**) Coefficients of GLMM fit for single trial ΔF/F during the secondary response window in each AldC compartment during licking trials by lick rate of the first lick bout in the trial (*n* = 5, 6, 10, 12, 12, 10, 3, and two mice; 225, 346, 384, 621, 722, 586, 353, and 131 trials for 7+, 6−, 6+, 5−, 5+, 5a−, 5a+, and 4b− compartments, respectively). Colored dots and error bars represent mean ± s.e.m. Red and blue colors correspond to AldC+ and AldC− compartments, respectively. (**A, B**) The secondary response window is indicated by gray vertical lines. A dashed vertical line represents the cue onset. Thick lines and shadings represent means ± s.e.m. (**B**) One-way ANOVA with repeated measures: *p<0.05. (**C**) ANOVA on GLMM fit: *p<0.05, **p<0.01.

DOI: https://doi.org/10.7554/eLife.47021.029

The following source data is available for figure 6:

**Source data 1.** Datasets used to create *Figure 6*.

DOI: https://doi.org/10.7554/eLife.47021.030

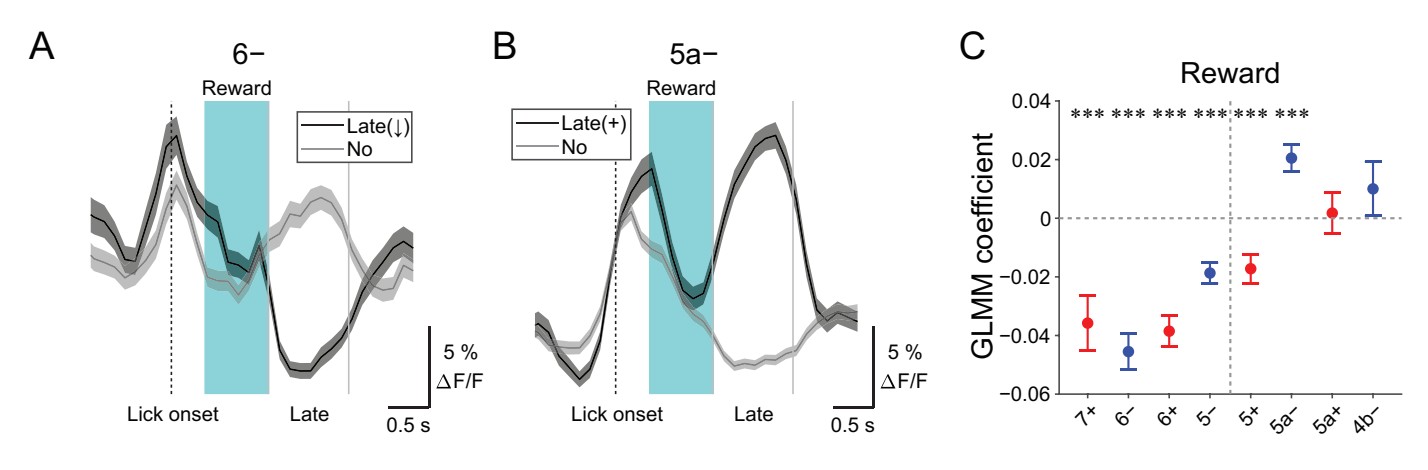

**Figure 7.** Late changes in CF signals represent positive reward outcomes. (A) Single trial calcium traces in a 6− compartment with (late (↓); black) or without (no; gray) decreases in CF signals during the late response window aligned to lick onset during licking trials (hit and FA) and non-task licking epochs (1.2 to 2.2 s from lick onset). (B) Same as panel (A) but for increases (late (+)) in a 5a− compartment. (C) Coefficients of GLMM fit for single trial ΔF/F during the late response window in each AldC compartment during licking trials (hit and FA) and non-task licking epochs by the presence (1) or absence (0) of reward in the trial (n = 5, 6, 10, 12, 12, 10, 3, and two mice; 427, 627, 736, 1081, 1270, 1137, 675, and 263 trials and epochs for 7+, 6−, 6+, 5−, 5+, 5a−, 5a+ and 4b− compartments, respectively). Colored dots and error bars represent means ± s.e.m. Red and blue colors correspond to AldC + and AldC− compartments, respectively. ANOVA on GLMM fit: ***p<0.001. (A, B) The late response window is indicated by gray vertical lines. A dashed vertical line represents the lick onset. Light blue shadings represent the window for reward delivery. Thick lines and shadings represent means ± s.e.m.

DOI: https://doi.org/10.7554/eLife.47021.031

The following source data is available for figure 7:

**Source data 1.** Datasets used to create *Figure 7*.
DOI: https://doi.org/10.7554/eLife.47021.032

## Discussion

By combining a transgenic mouse line that allowed visualization of cerebellar modules, two-photon calcium imaging of population CF signals, and an auditory go/no-go task, we identified the spatial organization of CF signals across modules in the Crus II, which broadly corresponds to the expression of aldolase C/zebrin II and their mediolateral locations (*Figure 8*). CF signals in the AldC+ compartments show graded activity after sensory cues, reflecting the speed and duration of motor behavior, whereas those in the AldC− compartments show similar activity in response to sensory cues regardless of the presence and absence of motor behavior, and therefore represent a sensory response. CF signals in the lateral AldC compartments represent a sensory response, whereas those in the medial AldC compartments represent motor behavior. These spatially segregated CF signals represent the behavior of mice in a temporally organized manner: pre-cue decreases represent motor preparation; early increases represent motor initiation, timing, and speed, and go/no-go discrimination; secondary increases represent motor speed; and late decreases represent positive reward outcomes. These results constitute a comprehensive functional map of CF inputs to the Crus II (*Figure 8*) and provide insights into the region-specific contributions of the cerebellum to sensorimotor behavior.

### Anatomical correlates of functional differences in CF inputs

In the present work, we have shown that CF signals in anatomically defined modules are broadly tuned to distinct functions, and have emphasized the importance of anatomical connectivity in determining behavioral function. In accordance with our observation of diverse task-related CF signals across AldC compartments (*Figure 8*), lesions of inferior olivary subnuclei have been reported to cause specific behavioral deficits in cats (*Horn et al., 2010*). According to a grouping of cerebellar modules on the basis of a detailed tracing study (*Sugihara and Shinoda, 2004*), group I compartments (7+, 6+, and 5+) are innervated by CFs originating from the principal olive and neighboring

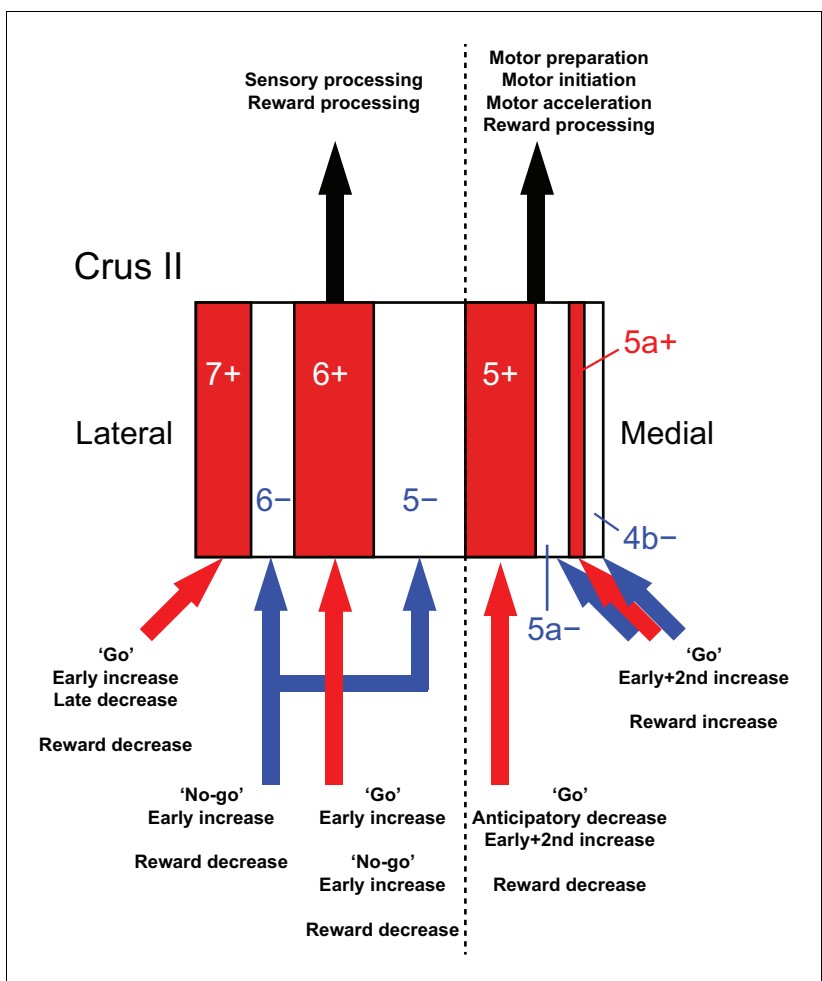

**Figure 8.** Diversity of CF signals across the entire Crus II and their functional implications. A schematic of our main findings. Red and white stripes represent AldC+ and AldC− compartments, respectively. A dashed line represents the functional boundary of lateral and medial Crus II. Red and blue arrows represent CF inputs to each compartment. Black arrows represent outputs from lateral and medial Crus II.
DOI: https://doi.org/10.7554/eLife.47021.033

areas. These compartments receive inputs from the cerebral cortex via the nuclei in the mesodiencephalic junction, including the red nucleus, area parafascicularis prerubralis, and nucleus of Darschewitsch (*Stuesse and Newman, 1990*; *Swenson and Castro, 1983*; *Veazey and Severin, 1982*). Furthermore, a recent study showed that perioral tactile information is directly conveyed by a pathway from the periphery to the Crus II via the mesodiencephalic junction and inferior olive (*Kubo et al., 2018*). Group IV compartments (6− and 5−) receive CF inputs from the dorsomedial group of the principal olive and the medial part of the dorsal accessory olive, which receive somatosensory inputs from the spinal cord, dorsal column nuclei, and trigeminal nucleus (*Molinari et al., 1996*; *Swenson and Castro, 1983*). Group II (5a+, 4b+, and 4+) and III compartments (5a− and 4b−) are innervated by CFs from the medial subnuclei and medial accessory olive, which receive somatosensory inputs from the spinal cord as well as vestibular and collicular inputs (*Akaike, 1992*; *Boesten and Voogd, 1975*; *Brown et al., 1977*; *Swenson and Castro, 1983*). Therefore, the CF inputs to group I (7+, 6+, and 5+) compartments mainly convey information from the cerebral cortex, while the majority of CF inputs to group IV and II/III compartments (6−, 5−, 5a−, 5a+, 4b−, 4b+, and 4+) send sensory information from the periphery. In terms of outputs, only PCs in the lateral part of group I and IV compartments (7+, 6−, and 6+) project to the dentate nucleus (*Sugihara, 2011*), which in turn projects back to a wide range of brain regions, including the motor and

association cortices (*Dum and Strick, 2003*). PCs located in the medial part of group I and IV compartments (5− and 5+) and those in group II/III compartments (5a−, 5a+, 4b−, 4b+, and 4+) project to the sensorimotor cortex via interpositus and fastigial nuclei, respectively (*Sugihara, 2011*). These differences in anatomical projections could account for the functional differences that we observed in CF inputs across AldC compartments.

## AldC modules provide a functional reference for CF inputs

Our analyses on the functional clustering of CF signals revealed broad congruence between the boundaries of the functional clusters and those of AldC expression (*Figure 2D* and *Figure 2—figure supplement 1*–4), indicating that the AldC profile can be used as an approximate for cerebellar functional modules at both cellular and microzonal resolutions. Previous studies have suggested several functional differences between AldC+ and AldC− PCs: (i) AldC+ PCs have a lower simple spike rate and a higher complex spike rate than AldC− PCs, which are dependent on co-expressing TRPC3 subtypes (*Zhou et al., 2014*); (ii) coupling between simple spikes and complex spikes is enhanced in AldC+ PCs (*Tang et al., 2017*); (iii) AldC+ PCs receive enhanced glutamate release by CF inputs when compared with AldC− PCs (*Paukert et al., 2010*), but show no difference in parallel fiber-PC synapses (*Nguyen-Minh et al., 2018*); (iv) at the population level, CF inputs to AldC+ and AldC− compartments are separately and precisely synchronized at the cellular resolution (*Tsutsumi et al., 2015*); and (v) AldC+ and AldC− PCs have been shown to respond differently to optic flow stimulation in the pigeon cerebellum (*Graham and Wylie, 2012*). In the present study, we found that more AldC+ PCs than AldC− PCs receive motor-related CF inputs (*Figure 2F*). Furthermore, the spatial organization of CF signals broadly corresponded to the AldC expression boundaries at both the microzone level (*Figure 2—figure supplement 1*) and the single cell level (*Figure 2—figure supplement 2*–4). This spatial clustering of similarly behaviorally relevant CF inputs benefits in terms of outputs because of an expected convergence of these densely clustered PCs on downstream neurons in the cerebellar nuclei, which exclusively read out synchronous inputs from these PCs (*Bengtsson et al., 2011*; *Person and Raman, 2012*; *Tang et al., 2019*). In addition to the CF system, future work on the differences in behaviorally relevant simple spike outputs from AldC+ and AldC− PCs will give us a comprehensive understanding of their complementary role in behavioral control.

## Mediolateral functional differences in CF inputs to the Crus II

Mediolateral difference in cerebellar function remains poorly characterized, although clear differences in anatomical projections (*Apps and Hawkes, 2009*; *Sugihara and Shinoda, 2004*; *Sugihara, 2011*) and functional couplings between the other brain areas (*Marek et al., 2018*) have been shown. In relation to AldC compartments, complex spike firing has been shown to be synchronized exclusively within AldC compartments in the lateral Crus II, whereas it can be synchronized across AldC compartments in the medial Crus II (*Sugihara et al., 2007*). These findings are further supported by our previous (*Tsutsumi et al., 2015*) and present findings (*Figure 2—figure supplement 2C*). However, behaviorally relevant CF signals across the entire mediolateral axis of a cerebellar lobule have never been investigated.

We found that CF signals in the medial Crus II were more representative of motor behavior, in that during hit trials, more CFs respond in the medial Crus II than in the lateral Crus II, and the responses are graded on the basis of lick rate (*Figure 3C and D*). Anticipatory decreases in CF signals in the medial Crus II were correlated with subsequent initiation of licking (*Figure 4B*), and early increases in these signals were correlated with lick initiation, presence of go cues, shorter lick latency, and faster licking (*Figure 5B,D and F*), suggesting their involvement in motor execution. Sharp peaks in CF signals in the medial Crus II at the onset of spontaneous licking (black lines in *Figure 4—figure supplement 3*) further support this notion. Moreover, secondary increases in CF inputs were observed almost exclusively in the medial Crus II (*Figure 4—figure supplement 1*) and they reflected faster licking during licking trials (*Figure 6C*), indicating their role in online control of licking.

By contrast, CF signals in the lateral Crus II were more representative of sensory responses, in that during CR trials, more CF signals were observed there than in the medial Crus II, and the responses were equally observed irrespective of licking or even pronounced after no-go cues

(*Figure 3C and D*), although movements such as jaw opening or tightening of the mouth specific to no-go cues could also contribute to these signals. Anticipatory decreases in CF signals in the lateral Crus II were correlated with delay in lick initiation (*Figure 4B*), which could be interpreted as a representation of lower attention to the task or as intention to delay the motor initiation (suppression of impulsive behavior). Early increases in these signals were correlated with the presence of no-go cues, but not with licking behavior (*Figure 5B,D and F*). Moreover, CF signals were sharply time-locked to sensory cues without a secondary rise during licking trials (*Figure 4—figure supplement 1*), together indicating their sensory nature.

These findings imply distinct behavioral functions of CF inputs to the lateral and medial cerebellum: CF inputs to the lateral Crus II might represent sensory saliency and/or attention signals, which could be conveyed via the dentate nucleus to be used for higher-order processing in the association cortices, whereas CF inputs to the medial Crus II could control motor behavior online (*Welsh et al., 1995*) by indirect projections, via the interpositus and fastigial nuclei, to either the motor cortex or brainstem motor neurons. Close monitoring of licking behavior (*Gaffield and Christie, 2017*) together with high temporal resolution imaging or targeted electrical recording from PCs will elucidate how the tongue trajectory is controlled by CF inputs to the medial Crus II. Sensory demanding task such as multi-sensory discrimination could further address the sensory function of CF signals in the lateral Crus II.

## Mechanisms of anticipatory decreases in CF inputs to the Crus II

We discovered a decrease in population CF signals for an upcoming sensory cue during the period when the mice were not licking (–1 to 0 s from cue onset; *Figure 4A and B*). In our task design, expert mice could predict the timing of cue presentation because the inter-trial interval was fixed at 6 s. We found that this anticipatory decrease in CF signals was associated with the subsequent initiation of licking (5+; *Figure 4B*), or with delay in lick initiation (5–; *Figure 4D*), which could reflect motor preparation, timing prediction, or shift in attention (engagement in the task). Given the anticipatory ramping-up activity in dentate nuclear neurons in monkeys representing timing prediction (*Ashmore and Sommer, 2013*; *Ohmae et al., 2017*), as well as fastigial/dentate nuclear neurons in mice representing motor preparation (*Chabrol et al., 2019*; *Gao et al., 2018*), the anticipatory decreases in CF signals that we observed could reflect ramping-up activity in these cerebellar nuclei via nucleo-olivary inhibitory projections (*Hesslow, 1986*). Simultaneous recordings from pairs of PCs and cerebellar nuclear neurons using highly efficient electrodes (*Jun et al., 2017*) will further elucidate the mechanism for how the decrease in CF signals contributes to anticipatory and preparatory activity in the cerebellum.

## Mechanisms of reward-related decreases in CF inputs to the Crus II

Positive reward outcomes were represented by decreases in CF signals at long latency (>1 s) in the lateral Crus II, and by increases in one of the medial AldC compartments (*Figure 7* and *Figure 4—figure supplement 3*). The reward outcome-related CF signals were consistent with recent reports (*Heffley et al., 2018*; *Heffley and Hull, 2019*; *Kostadinov et al., 2019*). Extremely long latency (~2.2 s from lick onset) of CF signals could be explained by longer duration of our reward delivery (0.41–1.23 s from lick; *Figure 7A and B*; see 'Materials and methods'). The lateral Crus II is reciprocally connected with the ventral tegmental area (*Ikai et al., 1992*). Furthermore, dopamine D1 receptors are expressed on a subset of neurons in the dentate nucleus, which has been shown to contribute to cognitive behavior (*Locke et al., 2018*). Therefore, it is possible that midbrain dopamine neurons send positive reward outcome-related information to the dentate nuclei, which in turn send inhibitory projections to the inferior olive, thus causing decreases in CF inputs to the lateral Crus II. This putative pathway could contribute to reward-based reinforcement learning of goal-directed behavior.

## Conclusions

Comprehensive monitoring of CF signals in the Crus II under the context of go/no-go discrimination behavior revealed the spatiotemporal organization of behaviorally relevant functions that broadly correspond to AldC expression and mediolateral locations. CF signals in AldC+ PCs represent motor behavior, whereas those in AldC– PCs represent sensory response. CF signals in the lateral Crus II

represent sensory aspects of the behavior, whereas those in the medial Crus II are correlated with motor execution. Together, temporally orchestrated activation of spatially organized CF inputs in the cerebellar hemisphere could be a basis for implementing sophisticated behavior.

# Materials and methods

## Key resources table

| Reagent type | Designation | Source | Identifier |
|---|---|---|---|
| Recombinant DNA reagent | pENN.AAV1. CMVs.PI.Cre.rBG | Addgene | RRID: Addgene_105537 |
| Recombinant DNA reagent | pAAV1.CAG.Flex. GCaMP6f.WPRE.SV40 | Addgene | RRID: Addgene_100835 |
| Genetic Reagent (*Mus musculus*) | C57BL/6NCrSlc | Japan SLC, Inc | RRID: MGI:5295404 |
| Genetic Reagent (*Mus musculus*) | Aldoc-tdTomato KI, C57BL/6N-Aldoc<tm1(tdTomato)Ksak> | RIKEN BioResource Center | MGI:6324252 |
| Software, algorithm | Matlab | MathWorks | RRID: SCR_001622 |
| Software, algorithm | Labview | National Instruments | RRID: SCR_014325 |
| Software, algorithm | Suite2P | GitHub | https://github.com/cortex-lab/Suite2P |
| Software, algorithm | Lick/no-lick task | GitHub | https://github.com/stsutsumi223/Aldoc-behavior |

## Animals

All experiments were approved by the Animal Experiment Committees of the University of Tokyo (#P08-015) and University of Yamanashi (#A27-1). Adult male heterozygous Aldoc-tdTomato mice (*n* = 17) (*Tsutsumi et al., 2015*) and adult male wild-type mice (Japan SLC, Inc, *n* = 5) at postnatal days 40–90 were used. The Aldoc-tdTomato mice did not show any noticeable behavioral phenotype. The mice were maintained on a reverse-phase 12 hr/12 hr light-dark cycle. All experiments were performed during the dark phase. Food was provided ad libitum but water intake was restricted to 1 mL/day during the task training. The Aldoc-tdTomato mouse line is available from the corresponding authors upon reasonable request, and the mice are also available at RIKEN BioResource Center (RBRC10927).

## Surgery and virus injection

Mice were anesthetized with isoflurane (5% for induction; 1.5–2.5% for maintenance). A mixture of drugs comprising the anti-inflammatory drug flunixin (2.5 mg/kg) and the antibiotics sulfadiazine (24 mg/kg) and trimethoprim (4.8 mg/kg) was injected intraperitoneally just before the skin incision. The exposed skull was covered with dental adhesive resin cement (Superbond; Sun Medical). A custom-made head plate (O'Hara and Co.) was fixed to the skull using photo-coagulative resin (Panabia; Kuraray). A 2.5 mm cranial window was created over the left Crus II (4 mm lateral from midline and 2 mm caudal from lambda). A pulled glass pipette (broken and beveled to an outer diameter of 30 μm) and a 5 μL Hamilton syringe were back-filled with a fluorine-based inert liquid (Fluorinert; 3M) and front-loaded with a virus solution. A mixture of adeno-associated viruses (total: 200 nL; pAAV1. CAG.Flex.GCaMP6f.WPRE.SV40 and pENN.AAV1.CMVs.PI.Cre.rBG at 1:1; Addgene; not diluted from original stock with $\geq 1 \times 10^{13}$ vg/mL) (*Najafi et al., 2014*) was injected directly into the left Crus II at a rate of 20 nL/min using a syringe pump (UMP3−1; WPI). The axis of the pipette was angled at 15° from the vertical axis and advanced by 270 μm from the pia along the axis of the pipette. After the injection, the pipette was kept in place for 5 min. The injection was performed twice at the medial and lateral parts of the Crus II. A 3 mm diameter glass coverslip (number 0 thickness; Matsunami Glass) was directly mounted over the dura, and its edge was sealed with tissue adhesive (Vetbond; 3M) and dental adhesive resin cement (ADFA; Shofu). After the surgery, mice were singly housed and allowed to recover for 8–14 days. Sulfadiazine/trimethoprim antibiotics were

included in the drinking water (12 g/L) during the recovery period. Abnormal behavior was not observed after AAV injections.

## Task training

For efficient training of mice to perform a lick/no-lick task under head fixation, we used a task training apparatus (O'Hara and Co.) similar to that described previously (*Isomura et al., 2009*). Water intake was restricted to 1 mL/day for 3 days before the onset of task training. Water was only provided from the lick port during task training. When mice failed to maintain 85% of their initial body weight after the task session, additional water (1–2 mL) was provided. Mice were trained once daily and typically underwent 3 days of pre-training sessions that consisted of two steps. First, mice were acclimated to a head-fixed configuration, and every lick on the lick port was rewarded with 2 μL of 0.1% saccharine solution (maximum: 2 μL/s) (*Isomura et al., 2013*). During the following 2 days, go cues (10 kHz tone) and no-go cues (4 kHz tone) were provided randomly. Only licks within 1 s after either cue were rewarded with 6 μL of saccharine solution (2 μL × 3; 0.41–1.23 s from lick onset; *Figure 7A and B*). The inter-trial interval was set at 6 s, but the subsequent trial was delayed by 1 s from the last lick if the mouse continued to lick. Once their success rate for the task exceeded 80%, the mice proceeded to a lick/no-lick task. In this task, licking of the port within 1 s after the go cue was rewarded with 6 μL of saccharine solution, whereas licking within 1 s after the no-go cue was punished with a time-out of 4.5 s. All tasks were controlled and recorded at 100 Hz ($n$ = 16 mice) or 200 Hz ($n$ = 1 mouse) by a custom-written program (*Tsutsumi, 2019*; copy archived at https://github.com/elifesciences-publications/Aldoc-behavior) using LabVIEW software (National Instruments). Licking behavior was monitored by the tongue crossing a line between infrared laser detectors (O'Hara and Co; *Figure 1A*).

## Muscimol injection

The $GABA_A$ receptor agonist muscimol (40–70 nL, 10 μg/μL) was pressure-injected directly into the left Crus II at the center of the cranial window of well-trained wild-type mice ($n$ = 5) using a glass capillary micropipette. At 30–60 min after muscimol injection, the lick/no-lick task session was started. The effect was observed throughout the session (~1 hr), but disappeared on the following day. For control experiments, mice were injected with the same amount of saline in the left Crus II on the following day, and started the task after 30–60 min.

## Behavior analysis

To evaluate the performance in the go/no-go task, we calculated the fraction of correct (hit and CR) trials and defined 80% as an expert performance (*Figure 1E*). For hit and FA trials, licking behavior within 4 s from the cue onset was analyzed (*Figure 1—figure supplements 1* and *2*). A lick bout was defined as a cluster of licks wherein the inter-lick interval was less than 1 s. Lick onset and offset were defined as the onset and offset of the first lick bout in each trial, respectively. Instantaneous lick rate was calculated as the inverse of the inter-lick interval and was assigned at each time point. The coefficient of variation (CV2), which was used to assess the regularity of licks (*Figure 1—figure supplements 1* and *2*), was calculated as described previously (*Holt et al., 1996*). Non-task licking epochs (nontask; *Figure 4—figure supplement 3*) were determined as a lick bout isolated from cues (at least 1 s before cues), meaning that the bout could not be associated with cues (spontaneous licking without a reward).

## Two-photon imaging

During the lick/no-lick task, two-photon imaging was performed using a two-photon microscope (MOM; Sutter Instruments) equipped with a 40 × objective lens (Olympus) controlled by ScanImage software (Vidrio Technologies) (*Pologruto et al., 2003*). Two-photon excitation was achieved with a pulsed Ti:sapphire laser (Mai Tai HP; Spectra-Physics) at a wavelength of 910 nm. The laser power was adjusted to ensure that it was below 40 mW at the sample to avoid phototoxicity. Fluorescence signals were divided into green and red channels with a dichroic mirror and emission filters (Chroma), and detected with GaAsP photomultiplier tubes (Hamamatsu Photonics). A pair of Galvano scanners was used for the first batch of mice ($n$ = 3), and a resonant scanner was used for the remaining mice ($n$ = 14). The imaging field was 180 × 230 μm at a depth of ~50 μm from the pia,

with a resolution of 128 × 128 pixels. Two thousand frames were acquired at 7.8125 Hz (256 s) for the first batch (*n* = 3), and 28,910 frames were acquired at 109.86 Hz with 14 frame averaging (net: 7.8471 Hz; 263 s) for the second batch (*n* = 13). For a separate mouse (*n* = 1), 28,770 frames were acquired at 107.18 Hz (268 s) without frame averaging.

First, tdTomato fluorescence on the red channel was imaged to identify the AldC compartments in the whole Crus II using the relative locations and widths of the compartments (*Sugihara and Quy, 2007*; *Tsutsumi et al., 2015*). Diagrams in *Figures 2A* and *3E* were created by manually tracing the boundary of tdTomato expression in a single animal. Typically, the 6+/5− boundary was at the center of the cranial window (4 mm lateral from midline and 2 mm caudal from lambda), and the relative locations of the 7+/6−, 6−/6+, 5−/5+, 5+/5a−, 5a−/5a+, and 5a+/4b− boundaries were 0.59 ± 0.05, 0.41 ± 0.04, −0.28 ± 0.05, −0.56 ± 0.05, −0.71 ± 0.09, and −0.80 ± 0.08 mm from the center of the window (*n* = total 10 mice; mean ± s.d.); corresponding to 4.59, 4.41, 3.72, 3.44, 3.29, and 3.20 mm lateral from midline, respectively.

Next, GCaMP6f calcium imaging was performed on the green channel at the boundaries of aldolase C expression. Before starting each imaging session, a tdTomato image on the red channel was obtained for reference (*Figures 2B* and *3A*, *Figure 2—figure supplements 2B* and *3A*, and *Figure 3—figure supplement 2A*). According to our functional separation of the Crus II, the 5−/5+ boundary (3.72 mm lateral and 2 mm caudal from lambda) delineates the lateral and medial Crus II (*Figure 3E* and *Figure 3—figure supplement 2C*).

## Imaging analysis

Imaging data was analyzed using MATLAB software (R2018a; MathWorks). To correct motion artifacts in the x-y plane and to extract regions of interest (ROIs) corresponding to PC dendrites, the Suite2P algorithm (*Pachitariu et al., 2017*) was used. Typical parameters were used for ROI extraction: Nk0 = 120 (number of clusters to start with), Nk = 60 (number of clusters to end with), sig = 0.5 (spatial smoothing length in pixels), and diameter = 2 (expected diameter of dendrites). After image registration and automatic ROI extraction, ROIs corresponding to PC dendrites were manually selected and non-PC-like ROIs were curated on the basis of the following criteria: (1) the shape of the ROI did not resemble that of a PC dendrite, being round or vessel-shaped for example; (2) a calcium trace generated by the mean pixel fluorescence in the ROI did not show a characteristic calcium increase caused by CF inputs (fast rise and slow decay); (3) calcium signals were saturated, such that an increase in fluorescence was sustained over a period in the order of seconds; and (4) the signal-to-noise ratio was too low (the lowest signal-to-noise ratio in our dataset was 5.68).

To reduce multiple counting of single PC dendrites as a result of oversegmentation, ROIs were merged if they satisfied the following criteria: (1) ROIs were adjacent or separated by two dendrites mediolaterally; and (2) the correlation coefficient between calcium traces from the ROIs was greater than 0.75. This process was iterated until no ROI pairs satisfied the criteria. After completion of these processing steps, imaging data that included at least five PCs on both sides of an AldC expression boundary obtained during an expert behavioral performance (fraction correct: >80%) were used for further analyses (total: 94 imaging sessions). The number of cells expressing GCaMP6f was counted on the basis of z-stacks obtained by two-photon imaging. We found that 53% of PCs (*n* = 5 stacks; 9.40 ± 0.51 cells/$10^4$ $\mu m^2$) and 6% of molecular layer interneurons (*n* = 5 stacks; 5.37 ± 3.44 cells/$10^6$ $\mu m^3$) expressed GCaMP6f, showing that molecular layer interneurons contributed only 5% of the total fluorescence signals at most.

## Event detection

ΔF/F was calculated from the raw fluorescence traces using the following equation: $(F − F_0) / (F_0 − F_b)$, where F is the raw fluorescence value, $F_0$ is the 8th percentile of the fluorescence values of the surrounding 12 imaging frames (−6 frame to +6 frame; total: 13 frames), and $F_b$ is the minimum fluorescence value of the mean image (*Ozden et al., 2012*). Response to the task was determined by an averaged ΔF/F trace for a given response window: pre-trial, −2 to −1 s from cue onset; pre-cue, −1 to 0 s from cue onset; early, 0 to 0.5 s from cue onset; and secondary, 0.5 to 1 s from cue onset (*Figure 4—figure supplement 1*). An additional response window was defined for lick bout onset-aligned traces: pre-lick, −2 to −1 s from lick onset; late, 1.2 to 2.2 s from lick onset (*Figure 4—figure supplement 3*). Responsive ROIs for each trial type for each response window (*Figures 2F* and

*3C* and D, *Figure 3—figure supplement 1*) were determined on the basis of trial-averaged ΔF/F within a response window exceeding mean + 2 s.d. of the baseline, which was defined as a ΔF/F trace that was less than the absolute Z-score of 1 calculated from the whole ΔF/F trace in an imaging session. For the calculation of response probability of single ROIs (*Figure 2—figure supplement 1*), a significantly positive response was determined on the basis that the single trial ΔF/F trace averaged within the early response window exceeded the mean + 2 s.d. of the baseline. Then the response probability was defined as the number of trials with the significantly positive response divided by the total number of trials for each trial type. Response of an AldC compartment was determined by using the same method, where a single trial population averaged ΔF/F trace was used instead of a single ROI ΔF/F trace. Mean + 2 s.d. and mean –0.5 s.d. values of the baseline ΔF/F trace of the AldC compartment were determined as thresholds to detect significantly positive and negative responses in each trial for each response window (*Figures 4–7* and *Figure 4—figure supplement 2*). For analyses of the pre-cue period, only trials without any appreciable licking (<1 Hz) during the pre-trial period were taken into account (*Figure 4* and *Figure 4—figure supplement 2*) in order to remove false positives resulting from differences in licking activities between these periods.

GCaMP6f-dependent $Ca^{2+}$ increases in PC dendrites evoked by CF inputs are shown to have ~33 ms in rise and ~100 ms in decay (*Gaffield et al., 2016*), which is fast enough to detect individual CF inputs (~1/ s). To ensure that the $Ca^{2+}$ signals that we observed correspond to complex spike firing in PCs, we detected $Ca^{2+}$ events during baseline (pre-trial period; −2 to −1 s from cue onset) by using local peak exceeding mean + 2 s.d. of whole trace (*Mukamel et al., 2009*). Baseline event rates were higher for AldC+ PCs than for AldC− PCs (n = 1295 and 1219 ROIs; 0.46 ± 0.02 and 0.43 ± 0.02 events/s; p=0.0012; paired t test), which was opposite to the data described in a previous report (*Zhou et al., 2014*). Absolute event rates were much lower than the reported values in awake mice (*Zhou et al., 2014*), which might be because of the poor temporal resolution of our imaging (7.8 frames/s): more than one complex spike could be included in the single detected event. Even if this is not the case, our result could be consistent with a homeostatic modulation of complex spike firing rate (*Ju et al., 2019*): frequent sensory stimulation causes a decrease in baseline complex spike firing rate.

## Generalized linear model with mixed effects (GLMM)

For the GLMM analyses in *Figures 4–7*, we fit a single trial population average ΔF/F in single AldC compartments for each response window by using one or two predictor variables as fixed effects and a mouse label as a random effect, as in the following equation:

$$\Delta F/F = \beta_0 + \beta_1 \times \text{fixed effect 1} + (\beta_2 \times \text{fixed effect 1}) + b$$

where β is coefficient of the fit (minimizing the difference between the model and the actual data), and b is a random effect (considering across mice variance). We used the MATLAB function fitglme to perform this calculation. We assumed normal distribution for the ΔF/F. Significance of coefficient for each fixed effect was determined by comparing model fit by removing that variable from the model.

## Statistics

All statistical analyses were performed using MATLAB software. All tests were two-tailed and significance was assigned to p=0.05. Data are represented as mean ± s.e.m. unless otherwise stated. Neither randomization nor blinding was performed in this study. Statistical significances and effect sizes for all the analyses are provided as *Supplementary file 1*.

## Acknowledgements

We thank Dr Mayumi Tada for her participation in the preliminary experiments, Dr Manabu Abe for the accession of the Aldoc-tdTomato mouse, and Drs Mitsuo Kawato and Keisuke Toyama, as well as members of the Kitamura and Kano laboratories, for helpful discussions. We also thank Alison Sherwin, PhD, from Edanz Group (www.edanzediting.com/ac) for editing a draft of this manuscript. This work was supported by JSPS KAKENHI (23115504, 25560432, 25113705, 25115705, 25290003,

15H01426, 17H03543, and 17H06313 to KK; 21220006, 25000015 and 18H04012 to MK); by Grants-in-Aid for Scientific Research on Innovative Areas (Comprehensive Brain Science Network and Brain Information Dynamics) from MEXT, Japan; by the Strategic Research Programme for Brain Sciences (Development of Biomarker Candidates for Social Behaviour) from MEXT, Japan (to MK); by Brain Mapping by Integrated Neurotechnologies for Disease Studies (Brain/MINDS) from AMED, Japan (to KK and MK); by the Takeda Science Foundation (to KK); and by the Uehara Memorial Foundation (to KK).

## Additional information

### Funding

| Funder | Grant reference number | Author |
|---|---|---|
| Japan Society for the Promotion of Science | 23115504 | Kazuo Kitamura |
| Ministry of Education, Culture, Sports, Science, and Technology | Comprehensive Brain Science Network | Kenji Sakimura Masanobu Kano Kazuo Kitamura |
| Ministry of Education, Culture, Sports, Science, and Technology | Brain Information Dynamics | Kazuo Kitamura |
| Ministry of Education, Culture, Sports, Science, and Technology | The Strategic Research Programme for Brain Sciences | Masanobu Kano |
| Japan Agency for Medical Research and Development | Brain/MINDS | Masanobu Kano Kazuo Kitamura |
| Takeda Science Foundation | | Kazuo Kitamura |
| Uehara Memorial Foundation | | Kazuo Kitamura |
| Japan Society for the Promotion of Science | 25560432 | Kazuo Kitamura |
| Japan Society for the Promotion of Science | 25113705 | Kazuo Kitamura |
| Japan Society for the Promotion of Science | 25115705 | Kazuo Kitamura |
| Japan Society for the Promotion of Science | 25290003 | Kazuo Kitamura |
| Japan Society for the Promotion of Science | 15H01426 | Kazuo Kitamura |
| Japan Society for the Promotion of Science | 17H03543 | Kazuo Kitamura |
| Japan Society for the Promotion of Science | 17H06313 | Kazuo Kitamura |
| Japan Society for the Promotion of Science | 21220006 | Masanobu Kano |
| Japan Society for the Promotion of Science | 25000015 | Masanobu Kano |
| Japan Society for the Promotion of Science | 18H04012 | Masanobu Kano |

The funders had no role in study design, data collection and interpretation, or the decision to submit the work for publication.

### Author contributions

Shinichiro Tsutsumi, Conceptualization, Data curation, Software, Formal analysis, Validation, Investigation, Visualization, Methodology, Writing—original draft, Project administration, Writing—

review and editing; Naoki Hidaka, Yoshikazu Isomura, Masanori Matsuzaki, Investigation, Methodology, Writing—review and editing; Kenji Sakimura, Resources, Methodology, Writing— review and editing; Masanobu Kano, Conceptualization, Supervision, Funding acquisition, Methodology, Writing—review and editing; Kazuo Kitamura, Conceptualization, Supervision, Funding acquisition, Validation, Investigation, Methodology, Writing—original draft, Project administration, Writing—review and editing

## Author ORCIDs

Shinichiro Tsutsumi ⬚ https://orcid.org/0000-0001-8926-2047
Masanori Matsuzaki ⬚ http://orcid.org/0000-0003-3872-4322
Masanobu Kano ⬚ https://orcid.org/0000-0002-0725-3292
Kazuo Kitamura ⬚ https://orcid.org/0000-0002-8956-4122

## Ethics

Animal experimentation: All experiments were approved by the Animal Experiment Committees of the University of Tokyo (#P08-015) and University of Yamanashi (#A27-1).

## Decision letter and Author response

Decision letter https://doi.org/10.7554/eLife.47021.037
Author response https://doi.org/10.7554/eLife.47021.038

## Additional files

### Supplementary files

• Supplementary file 1. Statistics results. Complete statistics results for all the figures.
DOI: https://doi.org/10.7554/eLife.47021.034

• Transparent reporting form DOI: https://doi.org/10.7554/eLife.47021.035

### Data availability

Data analysed for all the figures are included in the manuscript and source data files. The Aldoc-tdTomato mouse line is available at RIKEN Bio Resource Center (RBRC10927).

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
