## [Decision Letter]

Thank you for submitting your article "Modular organization of cerebellar climbing fiber inputs during goal-directed behavior" for consideration by *eLife*. Your article has been reviewed by Ronald Calabrese as the Senior Editor, a Reviewing Editor, and three reviewers. The reviewers have opted to remain anonymous.

The reviewers have discussed the reviews with one another and the Reviewing Editor has drafted this decision to help you prepare a revised submission.

Summary:

The reviewers agreed that the questions being addressed were interesting and the study has potential to be suitable for publication in *eLife*. However, there was concern that while the data are potentially very rich, the analysis and the spin chosen are rather weak. Although no additional experiments would be required, a substantial reanalysis of the data set is necessary, with proper statistics and data reporting with less pooling (allowing the reader to assess the variability of the responses by themselves).

In the consultation process there was general agreement about the strengths and weaknesses of the study. In particular, the following points were emphasized by all reviewers:

- All reviewers agreed that it would be interesting to know if there are sharp functional boundaries at zebrin transitions, but the manuscript in its current form does not achieve this, since the authors classified ROI's according to the zebrin stripe from the beginning based on the overall synchrony calculated from a whole imaging session. Specifically, all three Reviewers agreed that it would be important to establish the categories of dendritic responses independently from the zebrin and then show the correspondence between the two, i.e., analyze response type independent of anatomy.

- There is concern that the pooling by and across zebrin zone is poorly justified (the implicit logic in the manuscript is that there is synchrony within bands, hence, all Purkinje cells in single bands behave the same.) It seems that all CF-evoked calcium signals were averaged (across cells/animals, and even pooling across zones). This makes it difficult to assess the actual strength of the differences reported. (See reviewer 2's essential revision 3 and reviewer 1's essential revision 2.)

- Essential revisions 3 and 4 of reviewer 3 propose many possible analyses that the authors can try

- Orienting the manuscript toward the demonstration of the diversity of responses along a wide medio-lateral axis would fit well in *eLife*.

In preparing the revised manuscript, we ask you to please pay particular attention to these points, as well as to the individual comments of the three reviewers, which are appended below.

Reviewer #1:

The authors make use of a zebrin promoter-dependent transgenic mouse line to label microzones, thus allowing in vivo responses to be mapped to molecular markers. This is potentially an important contribution to understanding the functional mapping of cerebellum crus II.

Several concerns make it difficult to evaluate the paper in its current form. First, the mice are not generally available, which may not be in alignment with *eLife* policies. Second, claims are made for sharp boundaries between different zones, yet statistical analysis is not offered to support such a claim. The observations could also be consistent with gradients of responsiveness that do not correlate with boundaries. Third, other technical and biological claims are not adequately demonstrated by rigorous analysis.

No new experiments would be needed to address any of these comments. However, much more analysis is required.

Essential revisions

1): validation of anatomical locations

1) AldoII/tdTomato mice: Are these mice the same as the ones reported in Tsutsumi et al., 2015? That was four years ago. Are they now deposited at an internationally available resource? After they are deposited, in the Materials and methods section, describe where these mice came from and how others can gain access to them. Note that *eLife* journal policies state "Data, methods used in the analysis, and materials used to conduct the research must be clearly and precisely documented, and be maximally available to any researcher for purposes of reproducing the results or replicating the procedure." https://reviewer.elifesciences.org/author-guide/journal-policies

2) Locations: Although the emphasis is on the difference between zebrin-positive and zebrin-negative zones, the reward-vs-error signal seems to sort by anatomical position, not zebrin+/-. In this case, the zebrin bands act, in a sense, as a mediolateral coordinate system. What are the stereotactic coordinates of the zones in groups I, II/III, and IV? Is it possible to add such information to Figure 3H? Where is the distinction between medial and lateral Crus II? Please report the location of zebrin-band boundaries relative to visible landmarks. For instance: "The boundary between lateral bands (7+, 6+/-, and 5+/-, or Groups I and IV) and medial bands (5a+/- and 4b-, or Groups II and III) was found XXX +/- XXX μm (average+/-SD) lateral from the most medial visible location of crus II, its junction with STRUCTURENAME." This will maximize the impact of the authors' work, especially with other investigators who do not have the tdTomato line.

3) Figure 2A: Related to the comment above, explain how the diagram in Figure 2A was generated. Is this from a single animal or averaged? How reproducible are the compartments?

4) Figure 2B: It is hard to see expression patterns. Please show red and green images from two-photon microscopy side-by-side along with an overlay image.

5) Group II/III zones contain zebrin-positive and zebrin-negative zones, but this distinction is not made in the analyses for the group II/III zones. This is contrary to the analyses for groups I and IV, where different functions are described for CF in zebrin -positive and zebrin -negative zones. Figure 3—figure supplement 1A: looks like more CF activity following cue onset in the hit trials in the zebrin-positive zones. Given that based on these data, zones in group II/III are pooled, please show quantifications of CF activity following a cue in the zebrin-positive and zebrin-negative zones (not just as synchrony as in Figure 3—figure supplement 1C).

2: analysis of responses and qualitative claims

6) For several zones within a group, the claim is made that at the population level three different responses were found. That is, CF responses in group I zones during hit trials had 3 responses: (1) decrease during the pre-cue period. (2) early increase after the cue. (3) later decrease after the reward. Is anything known about these three different types of responses at the level of individual dendrites? Does one dendrite show all 3 types of responses, or can one distinguish a subset of the population only responsible for one of the responses?

7) Subsection “Difference in CF inputs between AldC+ and AldC− zones£”: "although these task-relevant CF responses were graded within zones, the boundary of the response pattern was largely congruent…": Explain this gradation in more detail. In what way and direction, were they graded? What is the quantitative distinction between a gradient and a sharp step?

8) Claims of segregation by microzone are not adequately supported. Show responses of PC dendrites based on location with 100-micron step. Also, please plot functional parameters for dendrites as a function of mediolateral distance from key microzone boundaries. If there is a microzone effect, it would be apparent as a step at the boundary.

9) Subsection “Mediolateral differences in CF responses”, "synchrony": define this term precisely so that readers do not have to look up another article to find out what was analyzed.

10) Subsection “Function of CF inputs 1 to group II/III zones”: "In summary, the population CF responses in group II/III zones tended to decrease before a cue,.…" (related to Figure 6). This is not apparent in Figure 6B. Also mentioned is a 'tendency' toward an anticipatory decrease (subsection “Function of CF inputs 1 to group II/III zones”). Where is the quantification for that? What is the size of this tendency? An effect size (i.e. difference divided by SD) would be helpful.

11) Figure 6—figure supplement 1, "higher temporal resolution images from these zones revealed rhythmic relationships with individual licks" (also subsection “Function of CF inputs 1 to group II/III zones”): What is the duration of individual licks? Is the temporal resolution of the calcium indicator high enough to capture responses to individual licks? Need to demonstrate.

12) How many imaging sessions were done per mouse? Figure 6—Figure supplement 1B and C: each trace is for 1 session. Are the 12 sessions in B and C the same 12 sessions as in A? Does that mean that each trace is from 1 mouse? Or is this whole figure an example from 1 mouse? Please clarify. If it is from one mouse, can we then also see data from averages of all mice to support this claim?

3: interpretation of results

13) Abstract: "These results indicate that spatially segregated CF inputs represent diverse brain functions, and are indispensable for execution of goal-directed behavior" This seems like an overstatement of the findings. The "diversity" of functions includes reward and motor initiation/termination, which would be "multiple functions." "Indispensable for execution of" clause is overstated and should be reworded to "may contribute to".

14) Introduction, "reward outcomes": Because complex spikes (CSs) could represent both rewards and errors, maybe change this to either "rewards" so it is clear that the CS encodes a positive reward event? Or "positive reward outcomes"

15) Discussion section: Discussion starts confusingly because the terminology changes. Is lateral zebrin+ the same as 7+/6+/5+ and Group I? Is lateral zebrin- the same as 6- and 5- or Group IV? And so on. I think the authors are trying to be clearer, but it is better to either introduce the lateral/medial concept earlier, or use all the terminology here so that readers can be anchored

16) Discussion section are confusing. I assume the inhibitory effect of the cerebellar nuclei on the inferior olive would explain how a decrease in population CF can reflect the ramping up activity in fastigial nuclei. Some mention of simple spike activity occurs in the Discussion section, but it would be helpful to include those earlier in the discussion, and explain the (hypothesized) mechanisms of how CF activity reflects ramping up activity in cerebellar nuclei with all other pathways involved as well.

Reviewer #2:

This study observed optic dendritic Ca^2+^ signal simultaneously from many Purkinje cells in identified anatomical compartments in crus II in mice during behavior performance of the go/no-go task. A combination of new techniques has been employed in this study. The main finding is that the type of response is well correlated with the anatomical compartment in which the Purkinje cell is located, and that the previous anatomical classification of compartments is well correlated with different type of responses. The results demonstrated functional importance of the anatomical compartmentalization of the cerebellum in animal behavior. The essential finding in results seems to be quite novel. Limitation of this study would be that (1) contents of the manuscript are basically descriptive about the recorded signal, without clear interpretation of mechanisms or consequences and that (2) time resolution of the signal seems poor to provide some time dependent aspects of CF inputs, such as the degree of synchronous firing of complex spikes among Purkinje cells (Sasaki, Bouwer and Llinas, 1989). Nevertheless, this report seems to have certain level of impact to general readers as well as researchers in the cerebellum field. The experiments seem to be performed with expertise. Writing is understandable. However, I have several concerns as follows.

1) The Ca^2+^ signal recorded in this study is not in parallel with the climbing fiber input (or complex spikes), in time or in intensity. Kitamura and Hausser, (2011), which have been cited in the manuscript, shows that (dendritic) Ca^2+^ signal decays in a slow time constant of ~1 s. On the other hand, the complex spike has a duration of several tens of ms and can fire repetitively at a frequency of ~10 Hz. Therefore, I think "CF input" in the manuscript should be replaced by "CF-dependent dendritic Ca^2+^" or something in many places. Also, the response speed (rise time, decay time, etc.) of protein Ca^2+^ indicator (GCaMP6f) used in experiments should be mentioned.

2) In the Abstract: "Crus II" should be mentioned, since "medial and/or lateral aldolase C-positive and/or -negative modules" cannot be defined unless the lobule is specified.

"…represent diverse brain functions". This study does not support this conclusion, since no "functions" were demonstrated. Instead, this study demonstrated different types of Ca^2+^ signals in relation to go/no-go tongue behavior.

"…are indispensable for.." This study does not support this conclusion. This phrase should be omitted.

3) Concerns about experiments and analyses:

"ROI" should be clearly defined. It seems that the authors assume that one ROI is equivalent with one Purkinje cell. But, in Figure 3A, the size of single ROIs varies significantly.

Basic information of experiments, for example, the number of ROIs (or Purkinje cells) in each compartment should be provided. For example, in Figure 2A, the recording area covers 4b- and 7+ according to this drawing. But, no data from Purkinje cells in 7+ or in 4b- are provided in the paper. However, the authors’ claims that Purkinje cells in 5+, 6+ and 7+ are similar (group I) and Purkinje cells in 4b-, 5a+ and 5a- are similar (group II/III).

I could not find data that show that Purkinje cells in 5+, 6+ and 7+ (or Purkinje cells in 5- and 6-) show similar response patterns in this paper, although it is claimed.

Is it possible to quantify the Ca^2+^ response? For example, traces in Figure 2C,D. show signals of various intensity. How these signal correspond to or reflect complex spike firing frequencies?

Reviewer #3:

In this manuscript Shinichiro Tsutsumi and co-authors explore the dynamics of climbing fiber activity in the cerebellar cortex of head-fixed mice performing a simple auditory-cued go/no-go task. The climbing fibers (CF) are recorded by calcium imaging using two-photon microscopy in a perio-oral lobule of the cerebellum, and their functional identity is provided by their relation to the zebrin positive/negative bands of the cerebellar cortex. The main conclusion of the study is that CF innervating Purkinje cells belonging to different bands have different relation to behavior, with somewhat different modulation of their activity in relation to movement, movement omission, movement initiation, reward. Overall the experiments and data curation seem carefully performed but the statistical analysis and presentation of the results is currently unsatisfactory and makes the assessment of the strength of the author's claim difficult. Moreover, the differences seem in many cases more quantitative than qualitative, preventing firm conclusions to be drawn on a differential involvement of the cerebellar regions in behavior.

Essential revisions:

1) The relevance of the task chosen may be questioned; indeed, the inactivation of the cerebellar cortex seems mostly to decrease licking vigor/onset, yielding a moderate increase in correct rejection and decrease in hit rate. There is no real evidence that the animals fail to process the cue or cue-reward association very differently with or without cerebellum; the cerebellum could help thus to have a slightly better time-locking of licking (as already shown in Rahmati et al., 2014) the authors focus here on a signal associated with learning (CF discharge) but it is unclear what the cerebellum needs to learn in this task.

2) The task used by the authors does not include a delay between the cue onset and the availability of the water, making difficult to separate what is sensory representation, reward expectation, motor preparation and motor execution. For example, in many cases, the calcium responses are seen in the absence of lick (CR) (absence of calcium signal in "misses" could be due to lack of attention of the mouse); the 1 second time period before the cue is defined as a period of motor preparation but it could be as well caused by an attentional shift. It seems to me a number of experiments could have been attempted to examine their impact on the CF responses (changes in the protocol such as omission of the reward, delayed reward delivery, presentation of the sound in the absence of the lick port etc.).

3) The differences of activations seem in many cases quantitative more than qualitative. Is it possible (or even likely since microzones could be a few PC wide) that the functional segmentation by zebrin zones is too coarse: indeed inspection of the correlation matrices in Figure 2—figure supplement 1 and Figure 3—figure supplement 1 suggest the existence of sub-compartments in the lobules 5+, 5- and 4- (other zones are not illustrated). I wonder whether a more fruitful analysis would be instead to categorize single CF response using functional criteria (probability of firing correlated to licking vigor, latency, presence of a reward, etc.) and examine the spatial distribution of the cells which, based on the current analysis, should be constrained by the zebrin zonation. Finally, the lobule studied is known to receive sensory inputs from many regions of the oral/peri-oral area in the rat (although the C3/5- zone could be more associated with the forelimb); it may therefore be engaged in other motor responses than licking (e.g. tightening of the mouth in the case of active omission?); therefore interpreting an increased response as encoding motor suppression is questionable.

4) While the number of cells/animals seems appropriate, the statistics are questionable, and the analysis seems sometimes incomplete. First, many cases show highly significant p-values, but tiny effect sizes (<10% of the SD, see for example Figure 4—figure supplement 1H,I); it seems no repeated measure ANOVA was used throughout the paper. This has to be corrected and the effects should be tested by nesting the random effects of cells within mice. The ANOVA with Gaussian model is probably not adequate for testing proportions (which are bounded between 0 and 1). In many cases the Ns are not indicated (e.g. when splitting trials for presence/absence of early response). Second, the choice of data illustrated or analyzed may be aggregating multiple factors. To provide a few examples, the lick rate is sometimes presented just for cases with licks (Hit+FA) or for all cases (including trials without licks, which make little sense); in the latter case, the quantification of Miss would be more appropriate to illustrate the difference in behavior (e.g. Panels 4C and 4E, could be instead 4E + Miss rate histogram). Also, DF/F aggregates the CF response probability and the fluorescence amplitude of CF discharge, which may show considerable changes from trial to trial according to Figure 2BE and is not reported nor analyzed. It seems that two types of 'Early' CF response take place: either in the 0-.5 second following cue onset or in the next.5 second (in groups II-III-IV); it would be useful to analyze this phenomenon more in detail: is the absence of 0.5-1s CF response due to a high CF firing probability in the 0-0.5s period and an apparent refractory period of >.5 s? are the late 0.5-1s CF response due to a poor temporal tuning of the CF within single trial? or the ability of the CF to fire twice in that time window in the zones where this occurs? The synchrony in relation to the movement seems not explored (while it could help to enhance the contrast between types of responses).

[Editors' note: further revisions were requested prior to acceptance, as described below.]

Thank you for resubmitting your work entitled "Modular organization of cerebellar climbing fiber inputs during goal-directed behavior" for further consideration at *eLife*. Your revised article has been favorably evaluated by Ronald Calabrese (Senior Editor), a Reviewing Editor, and three reviewers.

The manuscript has been improved but there are some remaining issues that need to be addressed before acceptance, as outlined below:

There are still concerns about the extent to which the claims of segregation of responses by aldolase C/zebrin expression are supported by the data. However, we appreciate that the demonstration of a wide range of activity patterns that are now well documented by the glm and which are grossly -if not finely- delineated by the zebrin zones. We therefore ask you to revise the manuscript in order to de-emphasize the strong conclusion, that CF responses are strictly segregated by aldolase C banding, and emphasize instead the diversity of responses across regions more broadly. Specific examples of necessary changes are given by reviewers 1 and 3, below.

Reviewer #1:

The authors have addressed a number of concerns. I appreciate the availability of the mice.

However, a central claim, segregation of responses by aldolase C expression, is still not convincing. Therefore, I cannot recommend acceptance of this manuscript.

1) Subsection “AldC expression delineates task-related CF signals”: The choice of a silhouette threshold appears to be arbitrary. It is claimed that "Silhouette value was mostly above 0.7, indicating good separation." But inspection of Figure 2—figure supplement 2 shows that only 5-/5+ shows a clear segregation by correlation. The other boundaries are blurred.

Therefore their conclusion in subsection “AldC expression delineates task-related CF signals” is not suitable: "showing that unbiased clustering of ROIs based on their responses largely corresponds to separation using AldC expression, thus further verifying our approach to separately analyze function per each AldC compartment."

2) Figure 2—figure supplement 1: Similarly, quite often the probability in CF response varies within a microzone. 5a-5a+ doesn't show a difference in probability between the neighboring 100 microns of each zone, either for Hits, False Alarm, or Correct Rejection. Also, probabilities between individual imaging sessions (gray lines) are variable. So, their conclusion that "Significant differences in response probability were noted across all the AldC boundaries for either or all of hit, FA, and CR trials (Figure 2—figure supplement 1),.… " (subsection “AldC expression delineates task-related CF signals”) seems too far-fetched at the least. What does it say if all 4 zones of 100 μm differ from each other? The effect does not seem to always be AldC-dependent.

Reviewer #2:

This study observed optic dendritic Ca^2+^ signal simultaneously from many Purkinje cells in identified anatomical compartments in crus II in mice during behaviour performance of the go/no-go task. A combination of new techniques has been employed in this study. The main finding is that the type of response is well correlated with the anatomical compartment in which the Purkinje cell is located, and that the previous anatomical classification of compartments is well correlated with different type of responses. The results demonstrated functional importance of the anatomical compartmentalization of the cerebellum in animal behaviour. The essential finding in results seems to be quite novel. Limitation of this study is that (1) contents of the manuscript are basically descriptive about the recorded signal, without clear interpretation of mechanisms or consequences of findings (different types of CS signals in different compartments) and that (2) time resolution of the signal seems poor to provide some time dependent aspects of CF inputs. Nevertheless, this report seems to have certain level of impact to general readers as well as researchers in the cerebellum field. The experiments seem to be performed with expertise. Writing is understandable.

All concerns that I indicated in the previous version seem to be solved appropriately. The limitation of this study that I indicated in the previous manuscript still stays in the revised version of the manuscript. However, the authors are more conscious about the limitation of the study, and have corrected the manuscript accordingly. They omitted some conclusions that were not fully supported by the results in Abstract.

As a whole, the manuscript seems easier to understand and I do not have other comments. The authors also seem to have responded to comments of other reviewers, which other reviewers can judge better.

Reviewer #3:

This manuscript has been much improved and I feel the authors have satisfactorily addressed many of the concerns raised previously. I still have a few comments:

1) Match of boundaries of functional and anatomical zones: the phrasing in the text seems stronger than the figures. Microzones within zones seem to be visible in the data either using clustering (Figure 2—figure supplement 2) or distance (Figure 2—figure supplement 1). Notably, a number of clusters of dendrite are crossing the zebrin borders (Figure 2—figure supplement 2 panels 5a+/4b-, 6+/5-, 6-/6+). The quantitative arguments (Figure 2—figure supplement 4C) provides enough argument to group responses using anatomical boundaries in the rest of the paper, but I feel the presentation of the results should acknowledge more clearly that the zebrin zonation is far from covering all the functional segmentation.

2) The manuscript is much improved by the addition of all the data, but the message may become harder to grasp for the reader; a synthetic scheme might help to gather the main information currently distributed in many schemes and increase the impact of the manuscript.

[Editors' note: further revisions were requested prior to acceptance, as described below.]

Thank you for resubmitting your work entitled "Modular organization of cerebellar climbing fiber inputs during goal-directed behavior" for further consideration at *eLife*. Your revised article has been favorably evaluated by Ronald Calabrese (Senior Editor), a Reviewing Editor, and one reviewer.

The manuscript has been improved but there are some remaining issues that need to be addressed before acceptance, as outlined below:

In the previous decision letter, we had asked you to "revise the manuscript in order to de-emphasize the strong conclusion, that CF responses are strictly segregated by aldolase C banding, and emphasize instead the diversity of responses across regions more broadly." While the specific examples given by the reviewers have been attended to in the current revision, several problematic statements still remain, and the abstract in particular gives a misleading impression as to what is actually shown.

We ask you to please, read through the entire paper to make sure the concerns about the imperfect correspondence between functional and anatomical boundaries have been addressed. At a minimum, please address the following specific points:

1) The Abstract still overemphasizes the aspects of the study relating to the correspondence of the functional and anatomical boundaries. The abstract needs to be revised in accordance with the previous decision letter, and for clarity. We suggest the following changes:

- In the third sentence, please return to the original phrasing, so that the sentence ends with "to investigate the functional differences in CF inputs to modules." (at a minimum, remove "across aldolase C-positive and -negative")

- Please edit these sentences for clarity and readability: "Anticipatory decreases and early increases in CF signals in the medial modules represent quick motor initiation, and early and secondary increases in these modules represent fast motor behavior. Early increases in CF signals also discriminated go/no-go cues oppositely in the medial aldolase C-positive and lateral aldolase C-negative modules. Positive reward outcomes were indicated by decreases in CF signals in the lateral modules. " Perhaps it would be helpful to focus on stating each spatial region and describing the signals found there (as was the style of the original abstract). Please also tone down the conclusions regarding aldolase C patterning, as requested. It may be helpful use language such as, "broadly corresponding to molecular patterning" or similar, as used in previous reviews.

- Please revise the last sentence of the Abstract to reduce the emphasis on molecular segregation, either by removing "molecularly" or at least to moving it after "spatially" and qualifying it based on the concerns about the hard borders that have already been raised.

2) Additional problematic statements need to be addressed:

- Subsection “AldC expression delineates task-related CF signals”: 'indicating that the AldC expression boundaries delineate CF functions at microzone level to some extent.'

This is untrue as noted by the other reviewers as well. Maybe it would be more appropriate to say that AldC expression boundaries have a close (<100µm?) spatial correspondence with the CF functional boundaries?

- Subsection “AldC expression delineates task-related CF signals”: 'in many cases': this is again rather fuzzy. According to Figure 2—figure supplement 2, the coincidence of boundaries seemed to fit approximately at the cellular level (which is the nature of zebrin limits) for the 7+/6-, 5-/5+, 5a-/5a+ boundaries; the other boundaries seem far less perfect.

- Furthermore, in the Discussion section "spatial organization of CF signals was delineated at AldC expression boundary at both microzone level (Figure 2—figure supplement 1) and single cell level (Figure 2—figure supplement 2, Figure 2—figure supplement 3 and Figure 2—figure supplement 4)." Same comment as above; this is not supported by the figures. A statement cannot be sometimes true and sometimes wrong.

3) Figure 8: shouldn't "intermediate" (which is more spatial than temporal) become 'late'? It is unclear how the arrows to 5-/6- and 6+ are related to the text underneath.

---

## [Author Response]

Summary:The reviewers agreed that the questions being addressed were interesting and the study has potential to be suitable for publication in eLife. However, there was concern that while the data are potentially very rich, the analysis and the spin chosen are rather weak. Although no additional experiments would be required, a substantial reanalysis of the data set is necessary, with proper statistics and data reporting with less pooling (allowing the reader to assess the variability of the responses by themselves).

We appreciate very much for reviewers’ positive evaluation and constructive comments on our manuscript. We carefully reanalyzed the data as shown below to fully address the concerns.

In the consultation process there was general agreement about the strengths and weaknesses of the study. In particular, the following points were emphasized by all reviewers:- All reviewers agreed that it would be interesting to know if there are sharp functional boundaries at zebrin transitions, but the manuscript in its current form does not achieve this, since the authors classified ROI's according to the zebrin stripe from the beginning based on the overall synchrony calculated from a whole imaging session. Specifically, all three Reviewers agreed that it would be important to establish the categories of dendritic responses independently from the zebrin and then show the correspondence between the two, i.e., analyze response type independent of anatomy.

We addressed this point by adding the following two analyses:

1) Task-related climbing fiber (CF) response probability binned each 100 μm from Aldolase C (AldC) expression boundaries as suggested by reviewer #1, essential revision 2, point 8 (Figure 2—figure supplement 1; Results section).

2) Principal component analysis on the whole calcium trace followed by *k*-means clustering, irrespective of AldC expression (Figure 2—figure supplement 2, Figure 2—figure supplement 3 and Figure 2—figure supplement 4; Results section).

The results of these analyses clearly indicate that boundaries of functional clusters identified by dendritic responses correspond to anatomical boundaries determined by AldC expression.

- There is concern that the pooling by and across zebrin zone is poorly justified (the implicit logic in the MS is that there is synchrony within bands, hence, all Purkinje cells in single bands behave the same.) It seems that all CF-evoked calcium signals were averaged (across cells/animals, and even pooling across zones). This makes it difficult to assess the actual strength of the differences reported. (See reviewer 2's essential revision 3 and reviewer 1's essential revision 2.)

We justified pooling response by AldC compartment by high coincidence rate (>80%) between clusters identified by PCA and AldC expression (Figure 2 —figure supplement 2 and Figure 2 —figure supplement 4B and 4C), and by looking at single region of interest (ROI) response averaged across trials, rather than looking at ROI-averaged trace per single trial (Figure 2D, thin lines in Figure 2E, Figure 3B, and Figure 3—figure supplement 1B). Task-related CF signals were similar across ROIs within single AldC compartment.

Then we pooled response within compartment and sorted task-related responses in single compartments from lateral to medial (Figure 3—figure supplement 1C). Overall, the same AldC compartments shared task-related CF signals.

On the basis of these analyses, we have regrouped the compartments according to response properties and mediolateral axis (lateral AldC+; 7+ and 6+, lateral AldC–; 6– and 5–, medial AldC+; 5+ and 5a+, medial AldC–; 5a– and 4b–, Figure 3E) rather than the grouping based on the climbing fiber innervation pattern (Group I – IV).

Further analyses for behavioral correspondence of CF signals were performed on individual compartments without pooling across compartments (Figure 4, Figure 5, Figure 6 and Figure 7).

- Essential revisions 3 and 4 of reviewer 3 propose many possible analyses that the authors can try.

We have addressed these points as answers to reviewer #3.

- Orienting the manuscript toward the demonstration of the diversity of responses along a wide medio-lateral axis would fit well in eLife.

We have emphasized this point by scrutinizing task-related CF signals in each compartment along entire mediolateral axis (Figure 3—figure supplement 1C) and their behavioral relevance by using generalized linear model with mixed effects per compartment (Figure 4, Figure 5, Figure 6 and Figure 7).

Reviewer #1:Several concerns make it difficult to evaluate the paper in its current form. First, the mice are not generally available, which may not be in alignment with eLife policies. Second, claims are made for sharp boundaries between different zones, yet statistical analysis is not offered to support such a claim. The observations could also be consistent with gradients of responsiveness that do not correlate with boundaries. Third, other technical and biological claims are not adequately demonstrated by rigorous analysis.

In order to fully address the reviewer’s points, we have added substantial amount of analyses and improved the manuscript to explain them in detail.

1) The mice used have been available upon request since we published 2015 paper (Tsutsumi et al., 2015). However, according to this reviewer’s suggestion, we will deposit the mice at RIKEN Bio Resource Center when the paper is published.

2) Following the reviewer’s advice, we added analyses on task-related CF response probability binned each 100 μm from Aldolase C (AldC) expression boundaries (Figure 2—figure supplement 1), and showed that functional boundaries correspond to AldC expression boundaries. To look at these relationships at single cell level, we added principal component analysis (PCA) on the whole calcium trace followed by *k*-means clustering, irrespective of AldC expression (Figure 2—figure supplement 2–4), and showed that unbiased functional classification of CF signals mostly correspond to separation using AldC expression.

3) We have reanalyzed spatiotemporal pattern of CF responses at single dendrites and single compartments to avoid excessive pooling of the data. In particular, we added analyses using generalized linear model with mixed effects (GLMM) per compartment to closely look at behavioral relevance of single compartments controlled for across mice variations, rather than on pooled data.

We have performed proper statistics and the results are provided as Supplementary file 1.

No new experiments would be needed to address any of these comments. However, much more analysis is required.

We appreciate the reviewer’s constructive comment. We substantially added analyses to address all the points raised below.

Essential revisions1: validation of anatomical locations1) AldoII/tdTomato mice: Are these mice the same as the ones reported in Tsutsumi et al., 2015? That was four years ago. Are they now deposited at an internationally available resource? After they are deposited, in the Materials and methods section describe where these mice came from and how others can gain access to them. Note that eLife journal policies state "Data, methods used in the analysis, and materials used to conduct the research must be clearly and precisely documented, and be maximally available to any researcher for purposes of reproducing the results or replicating the procedure." https://reviewer.elifesciences.org/author-guide/journal-policies

The Aldoc-tdTomato mouse line have already been made available upon request since Tsutsumi et al., (2015) was published. However, following the reviewer’s advice, we are now preparing to deposit these mice at RIKEN Bio Resource Center and they would be available after this work will be published. We have added this information in Materials and methods section.

2) Locations: Although the emphasis is on the difference between zebrin-positive and zebrin-negative zones, the reward-vs-error signal seems to sort by anatomical position, not zebrin+/-. In this case, the zebrin bands act, in a sense, as a mediolateral coordinate system. What are the stereotactic coordinates of the zones in groups I, II/III, and IV? Is it possible to add such information to Figure 3H? Where is the distinction between medial and lateral Crus II? Please report the location of zebrin-band boundaries relative to visible landmarks. For instance: "The boundary between lateral bands (7+, 6+/-, and 5+/-, or Groups I and IV) and medial bands (5a+/- and 4b-, or Groups II and III) was found XXX +/- XXX μm (average+/-SD) lateral from the most medial visible location of crus II, its junction with STRUCTURENAME." This will maximize the impact of the authors' work, especially with other investigators who do not have the tdTomato line.

We manually mapped the relative location of these AldC bands on the basis of tdTomato fluorescence on two-photon imaging. Relative and absolute coordinates of all the AldC boundaries were measured with reference to the 6+/5− boundary located at center of the cranial window (2 mm lateral from the midline and 4 mm caudal from lambda) and we have added these values in Materials and methods section. We have also added a scale bar to the diagram in new Figure 3E on the basis of these measurements.

3) Figure 2A: Related to the comment above, explain how the diagram in Figure 2A was generated. Is this from a single animal or averaged? How reproducible are the compartments?

As above, the diagrams in Figure 2A and new Figure 3E were manually made on the basis of tdTomato fluorescence on two-photon imaging from a single animal. This information is now added to Materials and methods section. We have previously shown that the boundaries of tdTomato expression in these mice are exactly the same as those of AldC expression by using immunohistochemistry (Tsutsumi et al., 2015) (Results section), and the expression pattern of AldC in inbred mice should be the same.

4) Figure 2B: It is hard to see expression patterns. Please show red and green images from two-photon microscopy side-by-side along with an overlay image.

We thank the reviewer for this suggestion. In the new version of Figure 2B, we separately showed red channel (tdTomato), green channel (GCaMP6f), and ROI selection.

5) Group II/III zones contain zebrin-positive and zebrin-negative zones, but this distinction is not made in the analyses for the group II/III zones. This is contrary to the analyses for groups I and IV, where different functions are described for CF in zebrin -positive and zebrin -negative zones. Figure 3—figure supplement 1A: looks like more CF activity following cue onset in the hit trials in the zebrin-positive zones. Given that based on these data, zones in group II/III are pooled, please show quantifications of CF activity following a cue in the zebrin-positive and zebrin-negative zones (not just as synchrony as in Figure 3—figure supplement 1C).

Instead of pooling the data by grouping based on CF innervation pattern, we now showed task-related CF signals in each compartment sorted from lateral to medial (Figure 3—figure supplement 1C), and found some difference between AldC+ and AldC− compartments even in the medial Crus II, as the reviewer pointed out. Then we compared the responses in the lateral vs. medial Crus II separately for AldC+ and AldC− compartments (Figure 3C–D).

2: analysis of responses and qualitative claims6) For several zones within a group, the claim is made that at the population level three different responses were found. That is, CF responses in group I zones during hit trials had 3 responses: (1) decrease during the pre-cue period. (2) early increase after the cue. (3) later decrease after the reward. Is anything known about these three different types of responses at the level of individual dendrites? Does one dendrite show all 3 types of responses, or can one distinguish a subset of the population only responsible for one of the responses?

We thank the reviewer for raising this important issue. We analyzed single ROI, single trial traces (Figure 4—figure supplement 2) to address this point. Decreases in CF signals at pre-cue period (−1 to 0 s from cue onset), increases in CF signals at early (0 to 0.5 s from cue onset) and intermediate (0.5 to 1 s from cue onset) response window could be induced simultaneously even in single dendrites and single trial level (Figure 4—figure supplement 2A; Results section). Decreases in CF signals at late response window (1.2 to 2.2 s from lick onset; Figure 4—figure supplement 2B; Result section) were also observed in single dendrites.

7) Subsection “Difference in CF inputs between AldC+ and AldC− zones£”: "although these task-relevant CF responses were graded within zones, the boundary of the response pattern was largely congruent…": Explain this gradation in more detail. In what way and direction, were they graded? What is the quantitative distinction between a gradient and a sharp step?

We added analyses on task-related CF response probability binned each 100 μm from Aldolase C (AldC) expression boundaries (Figure 2—figure supplement 1) to address this point. We showed both sharp steps at the AldC expression boundaries (such as 5−/5+, hit trials) and gradients even within compartment (such as 6+/5−, hit trials).

8) Claims of segregation by microzone are not adequately supported. Show responses of PC dendrites based on location with 100-micron step. Also, please plot functional parameters for dendrites as a function of mediolateral distance from key microzone boundaries. If there is a microzone effect, it would be apparent as a step at the boundary.

According to this suggestion, as above, we added analyses on task-related CF response probability binned each 100 μm from aldolase C (AldC) expression boundaries (Figure 2—figure supplement 1) and found that task-related CF signals are delineated at AldC expression boundaries at microzone level.

9) Subsection “Mediolateral differences in CF responses”, "synchrony": define this term precisely so that readers do not have to look up another article to find out what was analyzed.

We refrain from using the term “synchrony” in the revised manuscript, because our imaging frame rate was only 7.8 frames/s, thus it is not adequate to discuss synchrony.

10) Subsection “Function of CF inputs 1 to group II/III zones”: "In summary, the population CF responses in group II/III zones tended to decrease before a cue,.…" (related to Figure 6). This is not apparent in Figure 6B. Also mentioned is a 'tendency' toward an anticipatory decrease (subsection “Function of CF inputs 1 to group II/III zones”). Where is the quantification for that? What is the size of this tendency? An effect size (i.e. difference divided by SD) would be helpful.

We now have these statistics per compartment (Figure 4—figure supplement 1 and Figure 4—figure supplement 3) and their effect sizes (Supplementary file 1). For example, pre-cue decreases in CF signals were only observed in a 5+ compartment (effect size = 0.18; Supplementary file 1; Results section). We now refrain from using the term “tended” in the revised manuscript.

11) Figure 6—figure supplement 1, "higher temporal resolution images from these zones revealed rhythmic relationships with individual licks" (also subsection “Function of CF inputs 1 to group II/III zones”): What is the duration of individual licks? Is the temporal resolution of the calcium indicator high enough to capture responses to individual licks? Need to demonstrate.12) How many imaging sessions were done per mouse? Figure 6—figure supplement 1B and C: each trace is for 1 session. Are the 12 sessions in B and C the same 12 sessions as in A? Does that mean that each trace is from 1 mouse? Or is this whole figure an example from 1 mouse? Please clarify. If it is from one mouse, can we then also see data from averages of all mice to support this claim?

Regarding both point 11) and 12), we removed the previous Figure 6—figure supplement 1 because it was only from one mouse, and therefore, it is difficult to claim this point.

3: interpretation of results13) Abstract: "These results indicate that spatially segregated CF inputs represent diverse brain functions, and are indispensable for execution of goal-directed behavior" This seems like an overstatement of the findings. The "diversity" of functions includes reward and motor initiation/termination, which would be "multiple functions." "Indispensable for execution of" clause is overstated and should be reworded to "may contribute to".

We agree that we overstated here, so now it is rephrased as “indicate that… play distinct roles for” (Abstract).

14) Introduction, "reward outcomes": Because complex spikes (CSs) could represent both rewards and errors, maybe change this to either "rewards" so it is clear that the CS encodes a positive reward event? Or "positive reward outcomes"

Thank you for pointing this out. It is rephrased as “positive reward outcomes” throughout the manuscript.

15) Discussion section: Discussion starts confusingly because the terminology changes. Is lateral zebrin+ the same as 7+/6+/5+ and Group I? Is lateral zebrin- the same as 6- and 5- or Group IV? And so on. I think the authors are trying to be clearer, but it is better to either introduce the lateral/medial concept earlier, or use all the terminology here so that readers can be anchored

We now listed up all the relevant AldC compartments for each grouping. Also we ceased to use the previous grouping as Group I–IV, and instead use simple grouping based on functional property and mediolateral location of each compartment: lateral AldC+, 7+ and 6+; lateral AldC−, 6− and 5−; medial AldC+, 5+ and 5a+; medial AldC−, 5a− and 4b− compartments (Figure 3C–E and Figure 3—figure supplement 2; Results section).

16) Discussion section are confusing. I assume the inhibitory effect of the cerebellar nuclei on the inferior olive would explain how a decrease in population CF can reflect the ramping up activity in fastigial nuclei. Some mention of simple spike activity occurs in the Discussion section, but it would be helpful to include those earlier in the discussion, and explain the (hypothesized) mechanisms of how CF activity reflects ramping up activity in cerebellar nuclei with all other pathways involved as well.

We are sorry for causing the confusions. It is still controversial how ramping up activity in cerebellar nuclei (both fastigial and dentate) is generated. According to this reviewer’s suggestion, we have removed discussion about simple spikes and only discuss the relationship between nuclear ramping up activity and decreases in CF signals (Discussion section).

Reviewer #2:This study observed optic dendritic Ca^2+^ signal simultaneously from many Purkinje cells in identified anatomical compartments in crus II in mice during behavior performance of the go/no-go task. A combination of new techniques has been employed in this study. […] The experiments seem to be performed with expertise. Writing is understandable. However, I have several concerns as follows.

We appreciate the reviewer’s evaluation of our work.

As for the limitations:

(1) We have added generalized linear model with mixed effects (GLMM) fitting trial-by-trial ΔF/F in each compartment to single trial behavior to address behavioral relevance of these climbing fiber (CF) signals.

(2) We agree that the temporal resolution of our experiments was too poor to address CF synchrony, thus the argument about synchrony is omitted from the revised manuscript.

1) The Ca^2+^ signal recorded in this study is not in parallel with the climbing fiber input (or complex spikes), in time or in intensity. Kitamura and Hausser, (2011), which have been cited in the manuscript, shows that (dendritic) Ca^2+^ signal decays in a slow time constant of ~1 s. On the other hand, the complex spike has a duration of several tens of ms and can fire repetitively at a frequency of ~10 Hz. Therefore, I think "CF input" in the manuscript should be replaced by "CF-dependent dendritic Ca^2+^" or something in many places. Also, the response speed (rise time, decay time, etc.) of protein Ca^2+^ indicator (GCaMP6f) used in experiments should be mentioned.

We agree that “CF input” is overstated for our study, because the temporal resolution of our imaging (7.8 frames/s) is not enough to quantify actual complex spikes, as the reviewer pointed out. We now rephrased it as “CF-dependent dendritic Ca^2+^ signals” or simply “CF signals” (Results section) throughout the manuscript. As for response speed of GCaMP6f, we cited Gaffield et al., (2016), where they showed that the rise and decay time constant are ~33 ms and ~100 ms, respectively, for CF-dependent Ca^2+^ signals in Purkinje cell dendrites. We have stated this in Materials and methods section.

2) In the Abstract: "Crus II" should be mentioned, since "medial and/or lateral aldolase C-positive and/or -negative modules" cannot be defined unless the lobule is specified.

Thank you for pointing this out. We have added “Crus II” in the Abstract.

"…represent diverse brain functions". This study does not support this conclusion, since no "functions" were demonstrated. Instead, this study demonstrated different types of Ca^2+^ signals in relation to go/no-go tongue behavior."…are indispensable for.." This study does not support this conclusion. This phrase should be omitted.

Regarding these points, we agree that we overstated here. We now describe the results as “indicate that […] play distinct roles for execution of goal-directed behavior” (Abstract).

3) Concerns about experiments and analyses:"ROI" should be clearly defined. It seems that the authors assume that one ROI is equivalent with one Purkinje cell. But, in Figure 3A, the size of single ROIs varies significantly.

We are sorry for the confusions. As in the Materials and methods section, we inspected the results of automatic ROI extraction and semi-automatically corrected by the criteria described (Materials and methods section). In brief, closely located fractured ROIs which showed highly correlated activity (r > 0.75) were regarded as dendrites of the same cell. Furthermore, there is a manual inclusion criterion for the shape of regions of interest (ROIs), where round or vessel-shaped ROIs were omitted (Materials and methods section). In a new Figure 2B, we separately showed selected ROIs to make this point clearer.

Basic information of experiments, for example, the number of ROIs (or Purkinje cells) in each compartment should be provided. For example, in Figure 2A, the recording area covers 4b- and 7+ according to this drawing. But, no data from Purkinje cells in 7+ or in 4b- are provided in the paper. However, the authors’ claims that Purkinje cells in 5+, 6+ and 7+ are similar (group I) and Purkinje cells in 4b-, 5a+ and 5a- are similar (group II/III).

We added the number of ROIs in each compartment in the main text (*n* = 472 compartments, 13.7 ± 5.7 ROIs, mean ± s.d.; Results section). We also added at least one representative data from all the aldolase C (AldC) expression boundaries (Figure 2 and Figure 3 and Figure 3—figure supplement 1). Similarity of task-related CF signals across compartments is now shown in Figure 3—figure supplement 1C. On the basis of this result, we determined the functional boundary for lateral and medial Crus II at 5−/5+ boundary (Results section).

I could not find data that show that Purkinje cells in 5+, 6+ and 7+ (or Purkinje cells in 5- and 6-) show similar response patterns in this paper, although it is claimed.

As above, similarity of task-related CF signals across compartments is now shown in Figure 3—figure supplement 1C. AldC compartments in the same division (lateral vs. medial, and AldC+ vs. AldC−) share task-related CF signals.

Is it possible to quantify the Ca^2+^ response? For example, traces in Figure 2C,D. show signals of various intensity. How these signals correspond to or reflect complex spike firing frequencies?

We might lack the temporal resolution to quantify single CF input as discussed above, nevertheless we tried to extract CF events by using a published method (Mukamel et al., 2009), where local peaks exceeding mean + 2 s.d. of whole trace were extracted as events. On the basis of this method, we found that baseline event rates (during pre-trial: −2 to −1 s from cue onset) were 0.46 ± 0.02 and 0.43 ± 0.02 events/s for AldC+ and AldC− PCs (*n* = 1,295 and 1,219 ROIs; Materials and methods section). Given that ΔF/F can represent both probability and amplitude of CF signals (Najafi et al., 2014), the event rate that we measured by simple thresholding could underestimate the actual complex spike rate (~1 spikes/s Zhou et al., 2014). Therefore, we used either binarized response for each region of interest (ROI) for each trial (Figure 2—figure supplement 1), or whole session (Figure 2 and Figure 3 and Figure 3—figure supplement 2), or relative change in ROI-averaged ΔF/F detected within single trials (Figure 4—figure supplement 1 and Figure 4—figure supplement 3).

Reviewer #3:In this manuscript Shinichiro Tsutsumi and co-authors explore the dynamics of climbing fiber activity in the cerebellar cortex of head-fixed mice performing a simple auditory-cued go/no-go task. […] Moreover, the differences seem in many cases more quantitative than qualitative, preventing firm conclusions to be drawn on a differential involvement of the cerebellar regions in behavior.

We appreciate the reviewer’s evaluation of our work. To address this reviewer’s concerns, we substantially added new analyses and statistics to make our conclusions stronger.

Essential revisions:1) The relevance of the task chosen may be questioned; indeed, the inactivation of the cerebellar cortex seems mostly to decrease licking vigor/onset, yielding a moderate increase in correct rejection and decrease in hit rate. There is no real evidence that the animals fail to process the cue or cue-reward association very differently with or without cerebellum; the cerebellum could help thus to have a slightly better time-locking of licking (as already shown in Rahmati et al., 2014); the authors focus here on a signal associated with learning (CF discharge) but it is unclear what the cerebellum needs to learn in this task.

We agree that we are not addressing the causal sensory role of the cerebellum in our task. Nevertheless, we still observed difference in CF signals in response to go and no-go cues, and they were opposite for the lateral and medial Crus II (Figure 5B). This result provides insights into spatial organization of sensory-related CF signals (lateral > medial) of the cerebellum.

We also agree that the role of the cerebellum (Crus II) in motor timing has been somehow shown and not novel (Welsh, 2002, and Rahmati et al., 2014, as pointed out). However, the present finding that motor timing is mostly relevant to CF signals in the medial Crus II rather than the lateral Crus II (new Figure 4, Figure 5 and Figure 6) is novel and we believe that our findings provides a new perspective on mediolateral differences in cerebellar function.

We are not claiming CF’s role in learning in the previous and revised manuscript, rather addressing their role in learned motor execution (Welsh et al., 1995, Welsh, 2002, and ten Brinke et al., 2017). Nevertheless, we totally agree that their role in goal-directed learning remains a very important topic and needs to be addressed in future. We added a sentence on this topic with regard to the reward-related CF signals (Discussion section).

2) The task used by the authors does not include a delay between the cue onset and the availability of the water, making difficult to separate what is sensory representation, reward expectation, motor preparation and motor execution. For example, in many cases, the calcium responses are seen in the absence of lick (CR) (absence of calcium signal in "misses" could be due to lack of attention of the mouse); the 1 second time period before the cue is defined as a period of motor preparation but it could be as well caused by an attentional shift. It seems to me a number of experiments could have been attempted to examine their impact on the CF responses (changes in the protocol such as omission of the reward, delayed reward delivery, presentation of the sound in the absence of the lick port etc.).

We agree that a delayed go/no-go task can clearly separate sensory, motor, and reward expectation-related signals. Nevertheless, we tried to separate out sensory and motor aspects of CF signals by building a generalized linear model with mixed effects (GLMM) using both the presence (1) and absence (0) of licking and the presence of go cue (1) and no-go cue (0) in the trial to fit single trial ROI-averaged ΔF/F of each compartment (Figure 5B). We found distinct contributions of sensory and motor variables to CF signals in each Aldolase C (AldC) compartments: lick initiation to 6−, 5+, and 5a− compartments (mostly medial Crus II), and go cues to 5+, 5a+, and 5a− (medial Crus II), and no-go cues to 6− and 5− (lateral AldC−), which suggested their distinct roles in sensory and motor functions (Results section; Discussion section).

As the reviewer pointed out, what we call “motor preparation” could also be explained by attentional shift or timing prediction. Actually, the observed delay in lick initiation after decreases in CF signals in the 5− compartment (Figure 4C–D) could be regarded as low level of attention to the task. We added these different interpretations in Discussion section.

We also agree that we did not address full potential of reward representations in CF signals (such as in Kostadinov et al., 2019). However, we believe that our findings of delayed (> 1 s) decreases in CF signals at the lateral Crus II and their increases in a medial compartment (Figure 4—figure supplement 3), and their significant representations of positive reward outcomes (Figure 7), could add another layer of (or task-specific) spatiotemporal representations of positive reward outcomes in CF signals (Discussion section).

3) The differences of activations seem in many cases quantitative more than qualitative. Is it possible (or even likely since microzones could be a few PC wide) that the functional segmentation by zebrin zones is too coarse: indeed inspection of the correlation matrices in Figure 2—figure supplement 1 and Figure 3—figure supplement 1 suggest the existence of sub-compartments in the lobules 5+, 5- and 4- (other zones are not illustrated). I wonder whether a more fruitful analysis would be instead to categorize single CF response using functional criteria (probability of firing correlated to licking vigor, latency, presence of a reward, etc.) and examine the spatial distribution of the cells which, based on the current analysis, should be constrained by the zebrin zonation. Finally, the lobule studied is known to receive sensory inputs from many regions of the oral/peri-oral area in the rat (although the C3/5- zone could be more associated with the forelimb); it may therefore be engaged in other motor responses than licking (e.g. tightening of the mouth in the case of active omission?); therefore interpreting an increased response as encoding motor suppression is questionable.

To fully address the functional separation irrespective of AldC expression identity, we performed principal component analysis on whole trace followed by *k*-means clustering to functionally separate CFs for each field of view (Figure 2—figure supplement 2). This analysis indeed revealed sub-compartmental structures: CFs in single fields of view were divided into three or more clusters in some cases, which showed distinct task-related responses (Figure 2—figure supplement 3). Nevertheless, this unbiased functional separation still largely corresponded to the separation using AldC expression boundary (coincidence rate > 80%; Figure 2—figure supplement 4), which validated our approach to group CFs on the basis of AldC compartments. We then performed the analyses regarding various functional aspects of CF signals on single AldC compartment basis instead of excessive grouping (Figure 4, Figure 5, Figure 6 and Figure 7), and showed functional differences between compartments along the entire mediolateral axis in Crus II.

We agree that the Crus II is also responsible for sensory processing and motor control other than licking, and also agree that CF signals during correct rejection trials could also be explained by non-monitored motor actions. We have added this possibility in Discussion section.

4) While the number of cells/animals seems appropriate, the statistics are questionable, and the analysis seems sometimes incomplete. […] The synchrony in relation to the movement seems not explored (while it could help to enhance the contrast between types of responses).

Effect sizes for all the comparisons are now summarized in Supplementary file 1(Materials and methods section). We agree that effect sizes are sometimes very small (< 0.2), in that case we attenuated the claims (Results section). We now use repeated measures ANOVA and GLMM to control for across mice variability in every single statistic in the figures. For data that has bounds such as percentage, response probability, and cluster numbers, we used ANOVA on ranks (Figure 2F, Figure 3C, 3D, Figure 2—figure supplement 1, Figure 2—figure supplement 4, and Figure 3—figure supplement 2). We also provided the number of trials and mice used for each analysis in the figure legends as well as in Supplementary file 1.

Lick rate analyses are now only using licking trials (hit and FA). We agree that ΔF/F can represent both probability and amplitude of CF signals (Najafi et al., 2014). Therefore we used either binarized response for each region of interest (ROI) for each trial (Figure 2—figure supplement 1), or whole session (Figure 2F, Figure 3C, 3D and Figure 3—figure supplement 2), or relative change in ROI-averaged ΔF/F was detected within single trials (Figure 4—figure supplement 1 and Figure 4—figure supplement 3).

We deeply appreciate this reviewer’s comment on the two phases of CF signals within “early” (0 to 1 s from cue onset) response window. Following the reviewer’s advice, we separated this “early” response window into two: 0 to 0.5 s (early) and 0.5 to 1 s (intermediate) from cue onset. We basically found the same results on the narrower early CF signals (0 to 0.5 s from cue onset; Figure 2, Figure 3 and Figure 5) as those of old broader early CF signals (0 to 1 s from cue onset), whereas we found clear differences across the lateral and medial Crus II with regard to the secondary increases in CF signals (0.5 to 1 s from cue onset; Figure 3—figure supplement 2) and their behavioral relevance (Figure 6), which were unclear in the previous version of our manuscript. Single ROI single trial traces (Figure 4—figure supplement 2A) suggest that even a single ROI can sometimes fire twice within 0–1 s time window.

We ceased to pursue synchrony of CF signals because we lack temporal resolution (frame rate of 7.8 frames/s) to support the claims.

[Editors' note: further revisions were requested prior to acceptance, as described below.]

[…] We therefore ask you to revise the manuscript in order to de-emphasize the strong conclusion, that CF responses are strictly segregated by aldolase C banding, and emphasize instead the diversity of responses across regions more broadly. Specific examples of necessary changes are given by reviewers 1 and 3, below.

We would like to thank the editors and reviewers for the second inspection and further comments on our manuscript. We appreciate their positive evaluation for our revised manuscript. We have addressed their additional concerns by correcting the corresponding figures (Figure 2—figure supplement 2A and new Figure 8) and texts.

We very much appreciate the reviewers’ positive evaluation on the changes made on our manuscript. We de-emphasized our results in our revised manuscript with regard to complete separation of CF functions by AldC expression. To further emphasize our main results on the diversity of CF signals, we added a summary figure for the readers to better understand the significance.

Reviewer #1:The authors have addressed a number of concerns. I appreciate the availability of the mice.

We appreciate the reviewer’s positive comment on the changes in our manuscript.

However, a central claim, segregation of responses by aldolase C expression, is still not convincing. Therefore, I cannot recommend acceptance of this manuscript.

We attenuated the statement that only some of the AldC boundaries delineate CF functions.

1) Subsection “AldC expression delineates task-related CF signals”: The choice of a silhouette threshold appears to be arbitrary. It is claimed that "Silhouette value was mostly above 0.7, indicating good separation." But inspection of Figure 2—figure supplement 2 shows that only 5-/5+ shows a clear segregation by correlation. The other boundaries are blurred.

We changed the expression regarding silhouette values (subsection “AldC expression delineates task-related CF signals”) and omitted the later part of the sentence that was pointed out, to avoid exaggeration of our results.

Therefore their conclusion in subsection “AldC expression delineates task-related CF signals” is not suitable: "showing that unbiased clustering of ROIs based on their responses largely corresponds to separation using AldC expression, thus further verifying our approach to separately analyze function per each AldC compartment."2) Figure 2—figure supplement 1: Similarly, quite often the probability in CF response varies within a microzone. 5a-5a+ doesn't show a difference in probability between the neighboring 100 microns of each zone, either for Hits, False Alarm, or Correct Rejection. Also, probabilities between individual imaging sessions (gray lines) are variable. So, their conclusion that "Significant differences in response probability were noted across all the AldC boundaries for either or all of hit, FA, and CR trials (Figure 2—figure supplement 1),.… " (subsection “AldC expression delineates task-related CF signals”) seems too far-fetched at the least. What does it say if all 4 zones of 100 μm differ from each other? The effect does not seem to always be AldC-dependent.

We appreciate the reviewer’s careful inspection of our data. We now attenuated the statement that only some of the AldC boundaries delineate microzones (subsection “AldC expression delineates task-related CF signals”).

Reviewer #2:[…] All concerns that I indicated in the previous version seem to be solved appropriately. The limitation of this study that I indicated in the previous manuscript still stays in the revised version of the manuscript. However, the authors are more conscious about the limitation of the study, and have corrected the manuscript accordingly. They omitted some conclusions that were not fully supported by the results in Abstract.As a whole, the manuscript seems easier to understand and I do not have other comments. The authors also seem to have responded to comments of other reviewers, which other reviewers can judge better.

We very much appreciate the reviewer’s positive evaluation on our substantial effort to improve our manuscript.

Reviewer #3:This manuscript has been much improved and I feel the authors have satisfactorily addressed many of the concerns raised previously. I still have a few comments:1) Match of boundaries of functional and anatomical zones: the phrasing in the text seems stronger than the figures. Microzones within zones seem to be visible in the data either using clustering (Figure 2—figure supplement 2) or distance (Figure 2—figure supplement 1). Notably, a number of clusters of dendrite are crossing the zebrin borders (Figure 2—figure supplement 2 panels 5a+/4b-, 6+/5-, 6-/6+). The quantitative arguments (Figure 2—figure supplement 4C) provides enough argument to group responses using anatomical boundaries in the rest of the paper, but I feel the presentation of the results should acknowledge more clearly that the zebrin zonation is far from covering all the functional segmentation.

We attenuated the statement that only some of the AldC boundaries delineate CF function (subsection “AldC expression delineates task-related CF signals”).

2) The manuscript is much improved by the addition of all the data, but the message may become harder to grasp for the reader; a synthetic scheme might help to gather the main information currently distributed in many schemes and increase the impact of the manuscript.

We appreciate this reviewer’s constructive comment. We added the summary schematic as a main figure (new Figure 8). We hope the addition increases the impact of our manuscript.

[Editors' note: further revisions were requested prior to acceptance, as described below.]

In the previous decision letter, we had asked you to "revise the manuscript in order to de-emphasize the strong conclusion, that CF responses are strictly segregated by aldolase C banding, and emphasize instead the diversity of responses across regions more broadly." While the specific examples given by the reviewers have been attended to in the current revision, several problematic statements still remain, and the abstract in particular gives a misleading impression as to what is actually shown.We ask you to please, read through the entire paper to make sure the concerns about the imperfect correspondence between functional and anatomical boundaries have been addressed. At a minimum, please address the following specific points:

We would like to thank the editors and reviewers for their comments on our revised manuscript. We apologize for insufficient revisions and problematic statements remaining in the previous version of the. According to their comments, we have corrected the Abstract, the main text, and the figures. We have reduced the strong emphasis on the correspondence between functional clusters and aldolase C expression throughout the manuscript.

1) The Abstract still overemphasizes the aspects of the study relating to the correspondence of the functional and anatomical boundaries. The abstract needs to be revised in accordance with the previous decision letter, and for clarity. We suggest the following changes:- In the third sentence, please return to the original phrasing, so that the sentence ends with "to investigate the functional differences in CF inputs to modules." (at a minimum, remove "across aldolase C-positive and -negative")

We have changed this sentence as suggested.

- Please edit these sentences for clarity and readability: "Anticipatory decreases and […] in the lateral modules. " Perhaps it would be helpful to focus on stating each spatial region and describing the signals found there (as was the style of the original abstract). Please also tone down the conclusions regarding aldolase C patterning, as requested. It may be helpful use language such as, "broadly corresponding to molecular patterning" or similar, as used in previous reviews.

According to these suggestions, we now describe the patterns of activity and their functions in medial and lateral modules respectively, instead of emphasizing correspondence with aldolase C expression. We also added a sentence as suggested. Abstract: The boundaries of CF functions broadly correspond to those of aldolase C patterning.

- Please revise the last sentence of the Abstract to reduce the emphasis on molecular segregation, either by removing "molecularly" or at least to moving it after "spatially" and qualifying it based on the concerns about the hard borders that have already been raised.

We have removed a word “molecularly” and changed the sentence as follows (Abstract); “These results indicate that spatially segregated CF inputs in different modules play distinct roles for execution of goal-directed behavior.”

2) Additional problematic statements need to be addressed:- Subsection “AldC expression delineates task-related CF signals”: 'indicating that the AldC expression boundaries delineate CF functions at microzone level to some extent.'This is untrue as noted by the other reviewers as well. Maybe it would be more appropriate to say that AldC expression boundaries have a close (<100µm?) spatial correspondence with the CF functional boundaries?

We have changed this sentence as suggested (subsection “Correspondence between task-related CF signals and AldC expression”).

The AldC expression boundaries have a close (≦ 100 µm) spatial correspondence with the CF functional boundaries.

- Subsection “AldC expression delineates task-related CF signals”: 'in many cases': this is again rather fuzzy. According to Figure 2—figure supplement 2, the coincidence of boundaries seemed to fit approximately at the cellular level (which is the nature of zebrin limits) for the 7+/6-, 5-/5+, 5a-/5a+ boundaries; the other boundaries seem far less perfect.

We have changed this part as follows (subsection “Correspondence between task-related CF signals and AldC expression”);

Mean coincidence rates across animals were as high as 90% in 7+/6−, 5−/5+, and 5+/5a− boundaries (Figure 2—figure supplement 4C), indicating that unbiased clustering of ROIs based on their responses corresponds to separation using AldC expression at cellular resolution in these boundaries, but more broadly tuned in the other boundaries.

- Furthermore, in the Discussion section "spatial organization of CF signals was delineated at AldC expression boundary at both microzone level (Figure 2—figure supplement 1) and single cell level (Figure 2—figure supplement 2, Figure 2—figure supplement 3 and Figure 2—figure supplement 4)." Same comment as above; this is not supported by the figures. A statement cannot be sometimes true and sometimes wrong.

We have changed this sentence as follows (subsection “AldC modules provide a functional reference for CF inputs”);

Furthermore, spatial organization of CF signals broadly corresponded to AldC expression boundaries at both microzone level (Figure 2—figure supplement 1) and single cell level (Figure 2—figure supplement 2, Figure 2—figure supplement 3 and Figure 2—figure supplement 4).

3) Figure 8: shouldn't "intermediate" (which is more spatial than temporal) become 'late'? It is unclear how the arrows to 5-/6- and 6+ are related to the text underneath.

We are very sorry for the confusion. We used the word “late” for the later time window (1.2 to 2.2 s from lick onset). Therefore, we now use “secondary” for this time window. We have changed Figure 8, and other corresponding figures and texts.

In addition, we have modified several sentences throughout the manuscript to de-emphasize the strong conclusions, corrected several minor errors and added a reference that was recently published (all modifications are in colored and underlined text).